# Pam16 and Pam18 were repurposed during *Trypanosoma brucei* evolution to regulate the replication of mitochondrial DNA

**Corinne von Känel**[1], **Philip Stettler**[1], **Carmela Esposito**[1], **Stephan Berger**[1], **Simona Amodeo**[1], **Silke Oeljeklaus**[2], **Salvatore Calderaro**[1], **Ignacio M. Durante**[3], **Vendula Rašková**[3,4¤], **Bettina Warscheid**[2]\*, **André Schneider**[1]\*

1 Department of Chemistry, Biochemistry and Pharmaceutical Sciences, University of Bern, Bern, Switzerland, 2 Faculty of Chemistry and Pharmacy, Biochemistry II, Theodor Boveri-Institute, University of Würzburg, Würzburg, Germany, 3 Institute of Parasitology, Biology Centre, České Budějovice, Czech Republic, 4 Faculty of Science, University of South Bohemia, České Budějovice, Czech Republic

¤ Current address: CEITEC–Central European Institute of Technology, Brno, Czech Republic
\* bettina.warscheid@uni-wuerzburg.de (BW); andre.schneider@unibe.ch (AS)

**Data Availability Statement:** The mass spectrometry proteomics data have been deposited to the ProteomeXchange Consortium via the PRIDE partner repository and are accessible using

## Abstract

Protein import and genome replication are essential processes for mitochondrial biogenesis and propagation. The J-domain proteins Pam16 and Pam18 regulate the presequence translocase of the mitochondrial inner membrane. In the protozoan *Trypanosoma brucei*, their counterparts are TbPam16 and TbPam18, which are essential for the procyclic form (PCF) of the parasite, though not involved in mitochondrial protein import. Here, we show that during evolution, the 2 proteins have been repurposed to regulate the replication of maxicircles within the intricate kDNA network, the most complex mitochondrial genome known. TbPam18 and TbPam16 have inactive J-domains suggesting a function independent of heat shock proteins. However, their single transmembrane domain is essential for function. Pulldown of TbPam16 identifies a putative client protein, termed MaRF11, the depletion of which causes the selective loss of maxicircles, akin to the effects observed for TbPam18 and TbPam16. Moreover, depletion of the mitochondrial proteasome results in increased levels of MaRF11. Thus, we have discovered a protein complex comprising TbPam18, TbPam16, and MaRF11, that controls maxicircle replication. We propose a working model in which the matrix protein MaRF11 functions downstream of the 2 integral inner membrane proteins TbPam18 and TbPam16. Moreover, we suggest that the levels of MaRF11 are controlled by the mitochondrial proteasome.

## Introduction

The parasitic protist *Trypanosoma brucei* has a unique mitochondrial biology. As in other eukaryotes, more than 95% of its mitochondrial proteins are encoded in the nucleus, synthesized in the cytosol, and imported into and across the mitochondrial membranes [1]. However, the trypanosomal mitochondrial protein import machineries show significant differences to

the dataset identifiers PXD046840 (TbPam16 SILAC RNAi data), PXD046845 (TbPam18 SILAC RNAi data), PXD046849 (TbPam16 SILAC CoIP data), and PXD053251 (MaRF11 SILAC RNAi data).

**Funding:** This study was supported in part by NCCR RNA & Disease, a National Centre of Competence in Research (grant number 205601 to A.S.) and by project grant SNF 205200 to A.S. both funded by the Swiss National Science Foundation (https://www.snf.ch/en). The funders had no role in study design, data collection and analysis, decision to publish, or preparation of the manuscript.

**Competing interests:** AS is a member of the PLOS Biology Editorial Board.

**Abbreviations:** BSF, bloodstream form; dKO, double knockout; FCS, fetal calf serum; gRNA, guide RNA; IM, inner membrane; kDNA, kinetoplast DNA; LC–MS, liquid chromatography–mass spectrometry; MCP, mitochondrial carrier protein; mHsp70, mitochondrial heat shock protein 70; MRP, mitoribosomal protein; MS, mass spectrometry; MTS, mitochondrial targeting sequence; mtRNAP, mitochondrial RNA polymerase; NYsm, New York single marker; ORF, open reading frame; OXPHOS, oxidative phosphorylation; PAM, presequence translocase-associated motor; PCF, procyclic form; sKO, single knockout; TAC, tripartite attachment complex; TMD, transmembrane domain; UTR, untranslated region; WB, western blot.

the prototypical systems of yeast and mammals [2–6]. The largest differences are found in the translocase of the inner mitochondrial membrane (IM), the TIM complex (S1A Fig). Essentially all eukaryotes have 2 TIM complexes, termed TIM22 and TIM23 [7,8]. The TIM22 complex mediates insertion of proteins into the IM that have multi-spanning membrane domains, such as mitochondrial carrier proteins (MCPs) [9,10]. The TIM23 complex imports presequence-containing proteins across or into the IM [11]. To import its substrates into the mitochondrial matrix, TIM23 associates with the matrix-exposed presequence translocase-associated motor (PAM). The PAM consists of 5 essential and highly conserved subunits [7]: the mitochondrial heat shock protein 70 (mHsp70) [12,13], its J-domain-containing co-factors Pam18 [14–16] and Pam16 [17], Tim44 [18] and the nucleotide exchange factor Mge1 [19–21].

In contrast, *T. brucei* has a single TIM complex only, which with minor compositional variations, imports both types of substrates [22]. Interestingly, the only trypanosomal TIM component sharing homology to a subunit of TIM complexes in yeast or mammals is TbTim17. TbTim17 is an orthologue of the Tim22 subunit of the TIM22 complex [7,8,23,24].

To import presequence-containing proteins, the trypanosomal TIM complex associates with a PAM module [42] containing the trypanosomal mHsp70 orthologue (TbmHsp70), which is essential for the import of presequence-containing proteins [25,26]. *T. brucei* contains bona fide orthologues of Pam18 and Pam16, termed TbPam18 and TbPam16. However, while they are essential for normal growth of procyclic form trypanosomes, they are not involved in mitochondrial protein import [26]. Instead, the function of Pam18, and likely Pam16, in the trypanosomal PAM is carried out by the non-orthologous, essential, J-domain-containing integral IM protein TbPam27 [26].

The evidence for this has been described in detail in a previous publication [26]. In short, TbPam27 is: (i) stably associated with the TIM complex; (ii) required for import of presequence-containing proteins in vivo; (iii) essential for the formation of an import-arrested presequence-containing but not for an import-arrested carrier substrate; and (iv) recovered in the pulldown of the import-arrested presequence-containing but not in the pulldown of the import-arrested carrier substrate. In contrast, TbPam18 and TbPam16 are not associated with any known constellation of the trypanosomal TIM complex. Furthermore, their depletion does not affect mitochondrial protein import nor the formation of the 2 import-arrested intermediates.

Based on these observations, an evolutionary scenario was proposed that aims to explain the transition from 2 ancestral TIM complexes, found in most eukaryotes, to the single TIM complex of trypanosomes [26]. It posits that in the ancestor of kinetoplastids, TbPam27 fortuitously interacted with TbTim17. This allowed mHsp70 to bind to the resulting TbTim17/TbPam27 complex forming a rudimentary PAM. As a consequence, the TbTim17-containing, TIM22-like TIM complex acquired the capability to import both, presequence-containing proteins and MCPs. Thus, the previously essential TIM23 complex became redundant and its subunits were lost. However, the proposed scenario cannot explain why TbPam18 and TbPam16 were retained during evolution and why they are essential for the growth of PCF *T. brucei* [26].

The single mitochondrion of trypanosomes contains a single nucleoid containing all mitochondrial DNA. This DNA forms the most complex mitochondrial genome known in nature and is termed kinetoplast DNA (kDNA) (S1B Fig). It consists of 2 types of DNA rings: maxicircles (ca. 25 copies, 23 kb each) and heterogenous minicircles (ca. 5,000 copies, 1 kb each), which are arranged in a large intercatenated network [27,28]. Maxicircles encode 16 subunits of the oxidative phosphorylation (OXPHOS) complexes, 2 mitoribosomal proteins (MRPs), and 2 rRNAs [27–29]. Twelve of their transcripts require RNA editing to become functional mRNAs. This process is mediated by small guide RNAs (gRNAs), which are the only genes encoded on the minicircles [30–33]. Minicircles are highly topologically interlocked and

comprise 90% of the kDNA network [34]. Maxicircles are also interlocked with each other [35], and in addition, interwoven into the minicircle network [36]. The resulting kDNA disk in the mitochondrial matrix is physically connected to the flagellum's basal body in the cytosol via the tripartite attachment complex (TAC) [37,38].

Minicircle replication begins with their release into the kinetoflagellar zone, located between the kDNA disk and the IM [39]. It occurs unidirectionally via theta structures [40]. Replicated minicircles migrate to the antipodal sites, which are protein complexes at opposing sites of the kDNA disk, where gaps are repaired and minicircles are reattached to the periphery of the network [41,42]. Maxicircles replicate like minicircles, but always remain interlocked with the kDNA. However, the details of the process and the factors required for it are not well understood [28,43]. Minicircle release and reattachment causes concentration of the catenated maxicircles in the center of the disk [35]. The concomitant replication and segregation of the kDNA network, mediated by the TAC and the basal bodies, results in the formation of a maxi-circle-containing filament between the 2 minicircles networks, termed Nabelschnur. Completion of kDNA segregation requires cleavage of this Nabelschnur, to unlink the daughter kDNAs [44,45].

Altogether, replication of the single kDNA network involves up to 150 different proteins and is tightly coordinated with the nuclear cell cycle [27,28]. However, the mechanism of this coordination is presently unknown. What has been shown is that the mitochondrial protea-some TbHslVU, composed of the 2 subunits TbHslV and TbHslU, acts a negative regulator of minicircle and maxicircle copy numbers and its depletion thus causes accumulation of giant kDNAs [46,47]. Intriguingly, up to date, only a single TbHslVU substrate has been identified, the maxicircle replication factor TbPIF2, whose levels are increased in TbHslVU depleted cells [47].

Here, we show that TbPam18 and TbPam16, while not involved in mitochondrial protein import, are required for the replication of the maxicircle component of the kDNA. Strikingly, this function is mediated by a soluble TbPam16-interacting protein whose levels appear to be controlled by TbHslVU.

## Results

### Depletion of TbPam18 or TbPam16 mainly affects MRPs and OXPHOS complexes

TbPam18 and TbPam16 are not required for protein import, but the fact that they are essential integral IM proteins indicates that they have another function linked to mitochondria [26]. To identify what this function might be, we quantified global changes in the mitochondrial prote-ome caused by the depletion of either of the 2 proteins. Previously established tetracycline-inducible TbPam18 and TbPam16 RNAi cell lines [26] were analyzed by stable isotope labeling by amino acids in cell culture (SILAC)-based quantitative mass spectrometry (MS). Surpris-ingly, neither TbPam18 nor TbPam16 were detected in the 2 SILAC RNAi experiments. How-ever, using a TbPam16 antibody, we found that after only 1 day of RNAi induction, TbPam16 levels were strongly reduced in both cell lines (Fig 1A). Thus, we conclude that (i) the stability of TbPam16 depends on TbPam18, in line with the idea that the 2 proteins form a heterodimer as in yeast; and (ii) that RNAi against TbPam16 is very efficient.

The 916 and 893 mitochondrial proteins [1,29,48] were detected in the SILAC RNAi data sets and the levels of 12% and 13% of them were reduced more than 1.5-fold in the TbPam18 and TbPam16 RNAi cell lines, respectively (Fig 1B). The most affected proteins included MRPs [29] of which 59% and 63% were depleted more than 1.5-fold in the 2 cell lines (Fig 1B, top panels). Furthermore, we found that 15% and 16% of all detected components of the

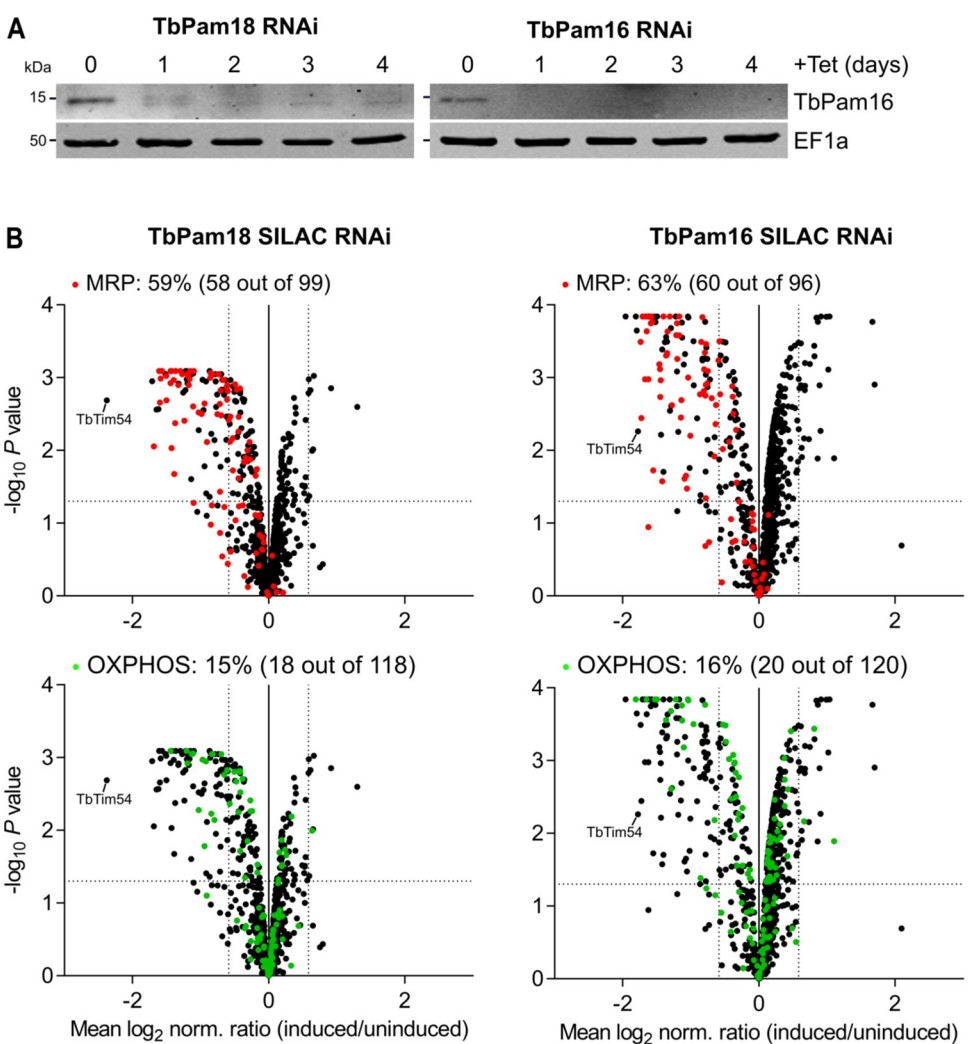

**Fig 1. TbPam18 and TbPam16 RNAi predominantly affects MRPs and OXPHOS components. (A)** Immunoblot analysis of steady-state protein levels of TbPam16 in whole-cell extracts of TbPam16 or TbPam18 RNAi cell lines over 4 days of induction. EF1a serves as loading control. **(B)** Global mitochondrial proteome changes upon ablation of TbPam18 (left panels) or TbPam16 (right panels). Mitochondria-enriched fractions of uninduced and 4 days induced TbPam18 and TbPam16 RNAi cells were analyzed by SILAC-based quantitative MS. Data sets were filtered for mitochondrial proteins and the mean $\log_2$ of normalized ratios (induced/uninduced) was plotted against the corresponding negative $\log_{10}$ of the adjusted *P* value (limma test). Highlighted are MRPs (red) and components of the OXPHOS pathway (green). The horizontal dotted line in each volcano plot marks an adjusted *P* value of 0.05. The vertical dotted lines indicate a fold-change in protein abundance of ±1.5. The percentages of all detected MRPs or OXPHOS proteins that are depleted more than 1.5-fold are indicated at the top of each panel. The number of all more than 1.5-fold depleted MRPs or OXPHOS proteins and the total number of all detected MRPs or OXPHOS proteins are shown in parentheses. Numerical data for panel (B) are available in S1 Table. MRP, mitoribosomal protein; MS, mass spectrometry; OXPHOS, oxidative phosphorylation.

OXPHOS pathway [48] were reduced more than 1.5-fold in the 2 cell lines (Fig 1B, bottom panels). In both experiments, complex IV was affected the most, followed by complexes I and III, whereas the levels of complex II and V subunits were not or only marginally decreased.

A common feature of the mitoribosome and the OXPHOS complexes I, III, and IV is that some of their subunits are encoded on the kDNA [27,29,49]. For mitoribosomes, these are the 12S and 9S rRNAs as well as 2 MRPs [29]. For complexes I, III, and IV, the number of maxicircle-encoded subunits is 6, 3, and 3, respectively. In contrast, only a single complex V subunit is

encoded on the kDNA and all complex II subunits are encoded in the nuclear genome [49]. In total, 67% (TbPam18) and 68% (TbPam16) of all more than 1.5-fold depleted mitochondrial proteins were either MRPs or subunits of the OXPHOS complexes I, III, IV, and V, all of which contain mitochondrially encoded RNAs and/or proteins (Fig 1B).

For protein categories consisting exclusively of nucleus-encoded proteins, the situation is very different. Neither of the 33 (TbPam18) and 30 (TbPam16) proteins detected in the category "kDNA replication factors" [43] were more than 1.5-fold depleted (S2A and S3A Figs). Essentially, the same results were obtained for 6 other major previously defined subgroups of mitochondrial proteins [48], which exclusively consist of nucleus-encoded proteins (S2B–S2F and S3B–S3F Figs). In all these subgroups combined, only 3 out of 279 detected proteins (TbPam18 RNAi) and not a single protein out of 274 (TbPam16 RNAi) were depleted more than 1.5-fold.

The observed phenotypes after depletion of TbPam18 or TbPam16 are notably different from RNAi cell lines targeting components of the protein import system, such as ATOM40 [1] or TbTim17 [50], in which depletion of members of all subgroups of mitochondrial proteins was observed. The SILAC RNAi results therefore confirm that TbPam18 and TbPam16 are not involved in mitochondrial protein import and suggest that the 2 proteins might be required for maintenance or replication of the kDNA.

## Depletion of TbPam18 and TbPam16 leads to shrinkage of the kDNA disk

To investigate the fate of the kDNA upon TbPam18 or TbPam16 depletion, we analyzed DAPI-stained RNAi cells by fluorescence microscopy. In line with the SILAC-RNAi analyses (Fig 1B), we found that after 4 days of RNAi induction, many TbPam18 and TbPam16 RNAi cells had smaller kDNAs compared to uninduced cells (Fig 2A, upper panels). A quantification of the experiments (Fig 2A, lower panel*s*) showed in both cell lines a time-dependent decrease of the kDNA size to 90% and about 60% after 3 (prior to the onset of the growth retardation) to 5 days of RNAi induction, respectively. Shrinkage of the kDNA disk has been observed previously when proteins involved in kDNA replication were ablated [47,51–55].

However, there is an alternative explanation. The most depleted protein upon TbPam18 RNAi and the fourth most depleted one upon TbPam16 RNAi was TbTim54 (Fig 1B). TbTim54 is unrelated to the yeast TIM22 complex subunit Tim54, but was proposed to mediate import of a subset of mitochondrial proteins with internal targeting sequences [56]. Thus, we wanted to exclude that the loss of maxicircles observed in the TbPam18 and TbPam16 RNAi cell lines could be a consequence of the co-depletion of TbTim54 (Fig 1B). Since RNAi depletion of TbTim54 did not affect growth [56], we produced a conditional double knock out cell line. In this cell line, depletion of the ectopically expressed copy of TbTim54 caused a growth arrest starting 4 days after tetracycline removal (S4A Fig). However, even after 6 days of tetracycline removal, no shrinkage of the kDNA was observed (S4B Fig). Thus, while TbTim54 is essential for normal growth, its depletion did not interfere with import of kDNA replication factors.

This is in line with the fact that even though TbTim54 has been reported to interact with TbTim17 [24], it was not enriched in any of our previously published pulldowns of TIM complex subunits or associated proteins (TbTim17, TbTim42, ACAD, TbTim13, TbPam27) [26,50,57]. These results, together with the observation that TbPam18 and TbPam16 are neither associated with the single trypanosomal TIM complex nor involved in mitochondrial protein import [26] (Figs 1B S2, and S3), suggest that TbPam18 and TbPam16 are more directly involved in kDNA replication or maintenance.

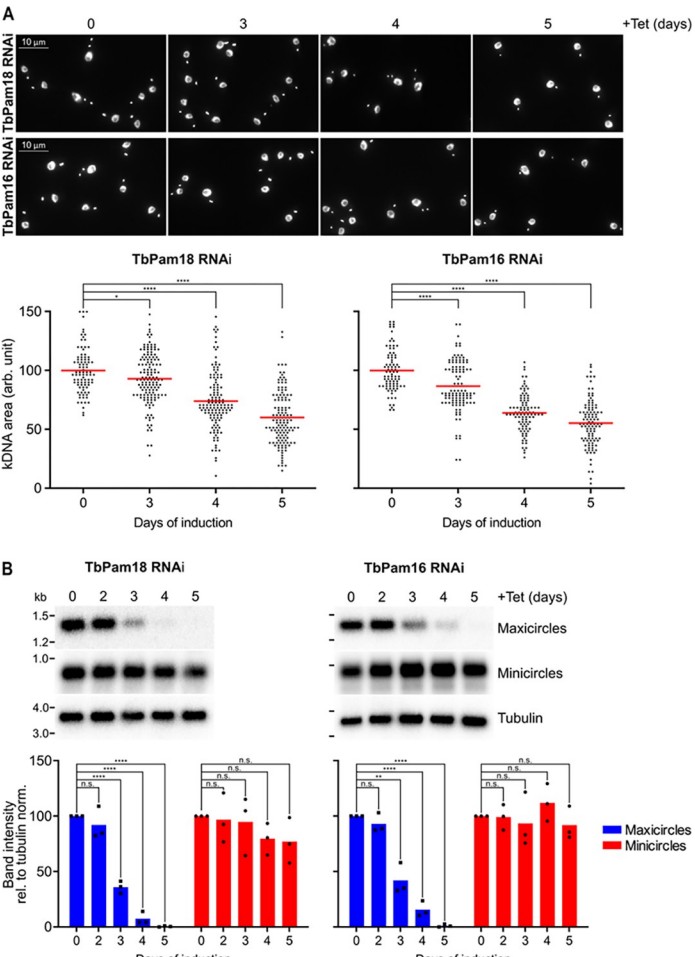

**Fig 2. TbPam18 and TbPam16 ablation causes the loss of maxicircles. (A)** Upper panels: Fluorescence microscopy analysis of DAPI-stained uninduced and 3 to 5 days induced TbPam18 and TbPam16 RNAi cells. Lower panels: Quantification of kDNA areas in 86 to 140 DAPI-stained RNAi cells induced for the indicated amount of time. The red line indicates the mean of the kDNA areas at each time point. The mean of the uninduced cells was set to 100%. *: *P* value <0.05, ****: *P* value <0.0001, as calculated by an unpaired two-tailed *t* test. **(B)** Southern blot analysis of steady-state levels of mini- and maxicircles in the TbPam18 and TbPam16 RNAi cell lines. Upper panels: Total DNA from uninduced or 3 to 5 days induced cells was isolated and digested with HindIII and XbaI. Probes specifically recognizing mini- or maxicircles were used. A probe detecting a 3.6-kb fragment of the tubulin intergenic region serves as loading control. Lower panels: Densitometric quantification of mini- and maxicircle abundance on Southern blots. The ratio of the mini- or maxicircle abundance and the respective loading control (tubulin) was normalized (norm.) to the ratios of uninduced cells. Blue (maxicircles) and red (minicircles) bars represent the mean of 3 independent biological replicates. n.s.: not significant, **: *P* value <0.01, ****: *P* value <0.0001, as calculated by an unpaired two-tailed *t* test. Numerical data for panels (A) and (B) are available in S1 Data. kDNA, kinetoplast DNA.

## Depletion of TbPam18 and TbPam16 leads to a selective loss of maxicircles

To study the effects of TbPam18 and TbPam16 depletion on the kDNA in more detail, we performed Southern blot analysis (Fig 2B). Total DNA was extracted from uninduced and induced RNAi cells, digested by restriction enzymes and separated on an agarose gel. The resulting blot was hybridized with mini- and maxicircle specific probes. Already after 3 days of RNAi induction, maxicircle levels were significantly reduced to about 39% in both cell lines. After 5 days, they were almost undetectable. In contrast, the levels of minicircles were not significantly changed over 5 days of RNAi induction. The same experiment (Fig 2B) was repeated

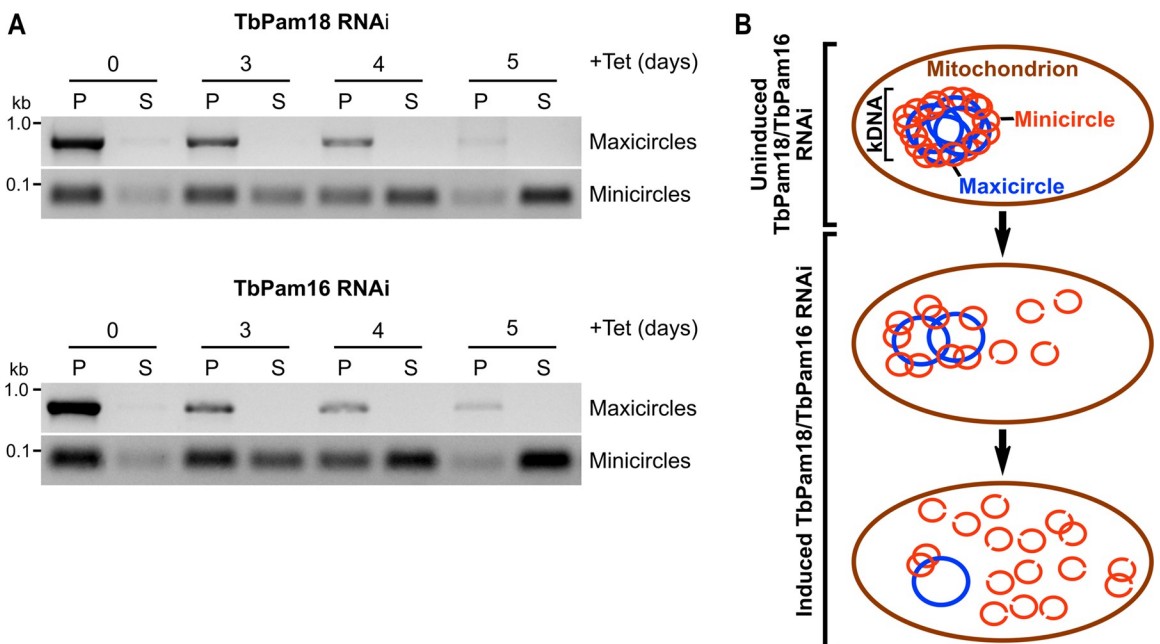

**Fig 3. Ablation of TbPam18 and TbPam16 causes accumulation of free minicircles. (A)** A quantitative PCR-based method was used to analyze steady-state levels of kDNA-bound or free mini- and maxicircles. Digitonin-extracted, mitochondria-enriched pellets from uninduced and 3 to 5 days induced TbPam18 and TbPam16 RNAi cells were solubilized in 1% digitonin. Subsequent centrifugation resulted in a pellet fraction (P) containing intact kDNA networks and a soluble fraction (S) containing free minicircles. DNA extracted from both fractions was used as template for PCR reactions amplifying specific mini- or maxicircle regions. PCR products were analyzed on agarose gels. **(B)** Schematic illustration of the putative sequence of effects on mini- and maxicircles upon RNAi-induced knockdown of TbPam18 and TbPam16. The ablation of TbPam18 and TbPam16 and the concomitant loss of maxicircles does not seem to inhibit the release of minicircles from the kDNA nor their replication. However, it appears to prevent the reattachment of free minicircles to the kDNA network. Consequently, free minicircles accumulate in the mitochondrial matrix. kDNA, kinetoplast DNA.

using PCR, to detect the changes in mini- and maxicircle levels and the same results were obtained (S5 Fig). The observation that the depletion of maxicircles is detected prior to the onsets of growth retardations (which occur at day 4) [26] suggests that TbPam18 and TbPam16 are directly involved in maxicircle replication or maintenance.

Since minicircles make up 90% of the kDNA [34], the massive network shrinkage seen in the DAPI stains of Fig 2A cannot be explained by a selective loss of maxicircles only [1,8]. One way to explain the constant levels of minicircles during TbPam18 and TbPam16 depletion (Fig 2B) is that they are released from the kDNA disk that progressively gets depleted from maxicircles.

To detect potential changes in free minicircle levels, digitonin-extracted, mitochondria-enriched pellets from uninduced and induced TbPam18 and TbPam16 RNAi cells were solubilized in 1% digitonin. Subsequent centrifugation resulted in pellets containing kDNA networks and supernatants containing free minicircles. PCR analysis of the DNA extracted from these fractions showed that maxicircles were only present in the pellets and that their levels decreased over time of induction as expected (Fig 3A). Minicircles behaved very differently. In uninduced cells, they were almost exclusively found in pellet fractions and thus in the kDNA networks. However, during the course of the RNAi, the amount of detected minicircles completely shifted to the supernatant (Fig 3A). Thus, ablation of TbPam18 and TbPam16, and the accompanying loss of maxicircles, does not inhibit the release of minicircles from the remaining kDNA network, nor their replication. But it appears to prevent their reattachment to the maxicircle-depleted networks (Fig 3B).

## TbPam18 and TbPam16 have procyclic form-specific functions

*T. brucei* has a complex life cycle alternating between an insect vector, the Tsetse fly, and a mammalian host. One of the replicative stages in the insect vector is the procyclic form (PCF), which contains an extensively reticulated mitochondrion that is capable of OXPHOS. The replicative stage in the mammalian host is the bloodstream form (BSF), which has a less reticulated mitochondrion that cannot perform OXPHOS [27,48]. The BSF produces its energy exclusively by glycolysis. But because the mitochondrial membrane potential in BSFs is maintained by the $F_1F_o$ ATP synthase working in reverse, a subunit of which is encoded on the kDNA, an intact kDNA network is essential not only in the PCF but also for the BSF [58,59].

It was therefore surprising that RNAi-mediated ablation of TbPam18 in the New York single marker (NYsm) BSF strain [60] did not affect growth (Fig 4A). However, the interpretation of this result is complicated, because even efficient RNAi never eliminates all mRNAs. It could be that the small amount of TbPam18 still present in these cells is sufficient for growth. To not run into the same issue with TbPam16, we established a TbPam16 NYsm double knockout (dKO) cell line, whose growth was indistinguishable from its parent cell line (Fig 4B). (For unknown reason, we were not able to produce a dKO cell line for TbPam18, not even a conditional one.)

In line with the lack of growth phenotypes there was no significant change in the size of the kDNA networks (Fig 4C) in neither the TbPam18-depleted RNAi cell line nor in the TbPam16 dKO cell line. Moreover, Southern blot analysis showed that the mean of mini- and maxicircle abundance was even higher in the 2 depleted cell lines; however, this change was not statistically significant (Fig 4D). This demonstrates that the function of TbPam16 is indeed specific for PCF trypanosomes and that its role in maxicircle replication is redundant in BSF cells or taken over by another protein. Moreover, based on the results of the TbPam18 RNAi cell line (Fig 4A) and the observation that TbPam16 and TbPam18, as Pam18 and Pam16 in yeast, likely form a heterodimer (Fig 1A), we conclude that the function of TbPam18 is also probably redundant in the BSF.

## Integral membrane localization of TbPam18 and TbPam16 is functionally relevant

TbPam18 and TbPam16 are integral IM proteins [26], which raises the question whether this feature is essential for their function. To find out, we expressed RNAi-resistant TbPam18 and TbPam16 variants lacking their predicted transmembrane domains (TMDs) [61], as well as their short IMS-exposed N-termini (4 aa for TbPam18 and 33 aa for TbPam16) (Fig 5A) [55]. This was achieved by ectopically expressing the RNAi-resistant variants of the 2 proteins under tetracycline control in the background of the corresponding RNAi cell lines. In the complemented cell lines tetracycline addition therefore simultaneously induces RNAi and expression of the RNAi-resistant variants of the 2 proteins. Note that whereas the TbPam16 variants could be C-terminally HA-tagged, in the case of TbPam18, both N- and C-terminal tags abolished the function of the protein (Figs 5A and S6A). Expression of full-length TbPam18 or TbPam16-HA in RNAi cells ablated for the corresponding endogenous proteins, restored normal growth as expected (Fig 5B and 5C, left panels). However, the same was not the case for the ΔN-TbPam18 and ΔN-TbPam16-HA variants that lack the predicted N-terminal TMDs (Fig 5B and 5C, right panels). Both truncated variants were N-terminally fused to the mitochondrial targeting sequence (MTS) of trypanosomal TbmHsp60 to ensure their import into mitochondria. Since the TbPam16-HA and ΔN-TbPam16-HA variants were tagged, their expression and import could be verified biochemically using digitonin extractions (S6B Fig). The abundance of ΔN-TbPam16-HA is approximately 1.5-fold higher than the one of the full-

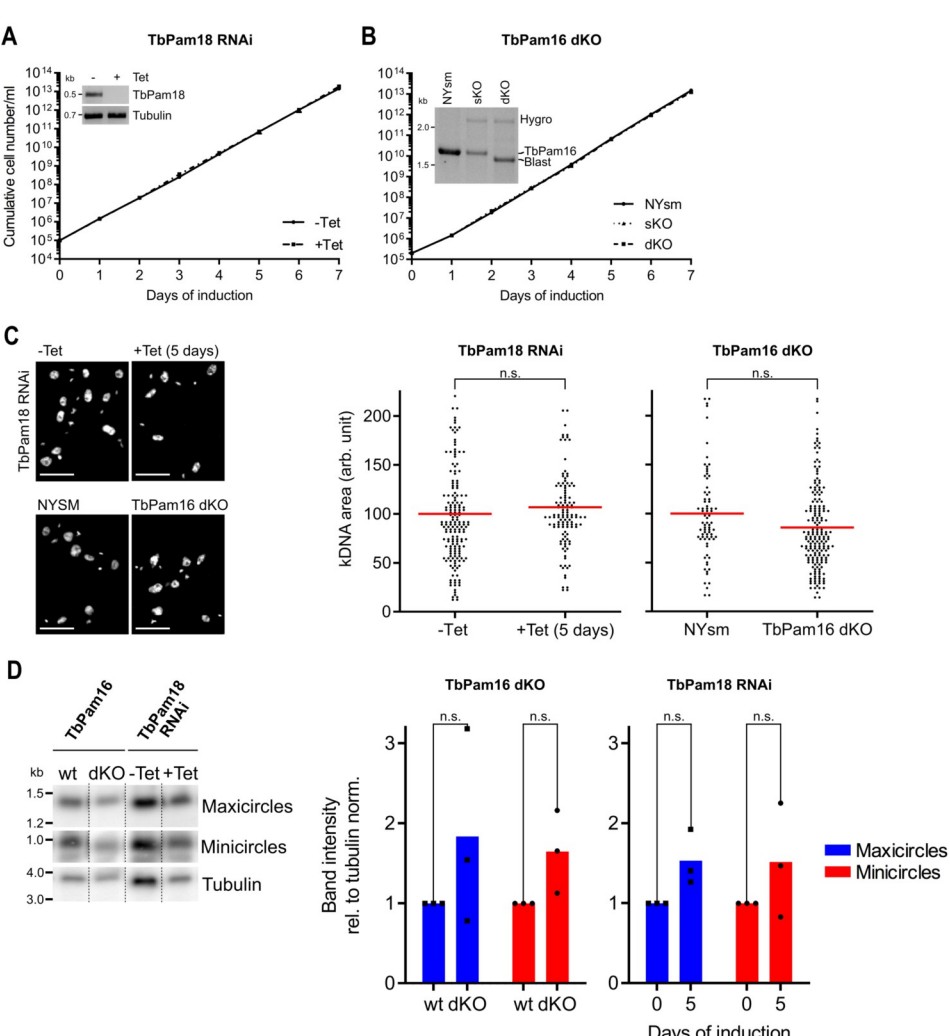

**Fig 4. TbPam18 and TbPam16 are not essential in BSF trypanosomes. (A)** Growth curve of uninduced (-Tet) and induced (+Tet) BSF NYsm RNAi cell line ablating TbPam18. Error bars correspond to the standard deviation (*n* = 3, error bars are too small to be visible). Inset: RT-PCR product of the wt TbPam18 mRNA in uninduced (-) or 2 days induced (+) cells. Tubulin mRNA serves as loading control. **(B)** Growth curve of NYsm, TbPam16 sKO, and dKO BSF cell lines. Inset: Verification of sKO and dKO by PCR using 1 primer pair to amplify the TbPam16 ORF (approximately 1.7 kilobases (kb)), the hygromycin (hygro, approximately 2.3 kb), or blasticidin (blast, approximately 1.6 kb) resistance cassettes at the same time. Hygro was used to replace the first allele and blast was used to replace the second allele. **(C)** Left: Comparison of DAPI-stained uninduced and 5 days induced BSF TbPam18 RNAi cells as well as BSF NYsm (parental cell line) and BSF TbPam16 dKO cells by fluorescence microscopy analysis. Right: Quantification of kDNA areas in 78 to 191 DAPI-stained cells. The red line indicates the mean of the kDNA areas. The mean of the uninduced TbPam18 RNAi cells and the NYsm cells was set to 100%. n.s.: not significant (*P* value >0.05) as calculated by a permutation test. **(D)** Southern blot analysis of mini- and maxicircles in NYsm (wt) cells, TbPam16 dKO cells as well as uninduced (-Tet) and 5 days induced (+Tet) BSF TbPam18 RNAi cells. Left: Total DNA from the indicated cell lines was isolated and digested with HindIII and XbaI. Probes specifically recognizing mini- or maxicircles were used. A probe detecting a 3.6-kb fragment of the tubulin intergenic region serves as loading control. Right: Densitometric quantification of mini- and maxicircle abundance on Southern blots. The ratio of the mini- or maxicircle abundance and the respective loading control (tubulin) was normalized (norm.) to the ratios of uninduced cells. Blue (maxicircles) and red (minicircles) bars represent the mean of 3 independent biological replicates. n.s.: not significant as calculated by an unpaired two-tailed *t* test. Numerical data for panels (A) to (D) are available in S1 Data. BSF, bloodstream form; dKO, double knockout; kDNA, kinetoplast DNA; NYsm, New York single marker; ORF, open reading frame; sKO, single knockout.

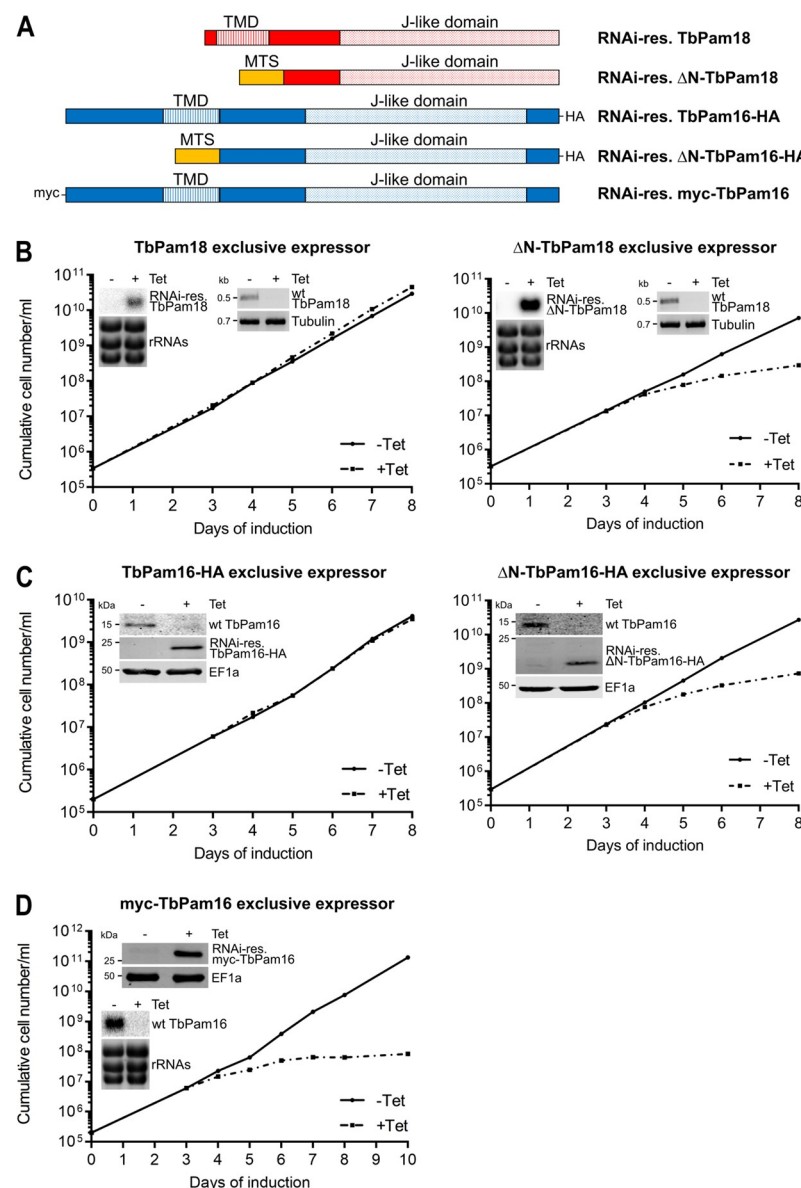

**Fig 5. Integral membrane localization of TbPam18 and TbPam16, as well as the IMS-exposed N-terminus of TbPam16 are functionally relevant. (A)** Schematic representation of RNAi-resistant (RNAi-res.) full-length and N-terminally truncated variants of TbPam18 and TbPam16. TbPam18 constructs are untagged, while TbPam16 constructs carry an N-terminal myc-tag or a C-terminal HA-tag. Predicted TMDs and J-like domains are indicated. Predicted TMDs and J-like domains are indicated. To ensure mitochondrial localization, the N-terminally truncated variants were expressed carrying the MTS of trypanosomal mitochondrial heat shock protein 60. **(B)** Growth curves of uninduced (-Tet) and induced (+Tet) cell lines ectopically expressing RNAi-res., full-length TbPam18 (left), or ΔN-TbPam18 (right) in the background of RNAi targeting the wild-type (wt) TbPam18 (TbPam18 and ΔN-TbPam18 exclusive expressors). Insets on the left: Northern blots of total RNA isolated from uninduced (-) and 2 days induced (+) cells probed for the mRNAs of RNAi-res. TbPam18 or ΔN-TbPam18 to confirm efficient inducible ectopic expression. Ethidium bromide-stained rRNAs serve as loading control. Insets on the right: RT-PCR products of the wt TbPam18 mRNA in uninduced (-) or 2 days induced (+) cells. Tubulin mRNA serves as loading control. **(C)** Growth curve of uninduced (-Tet) and induced (+Tet) cells ectopically expressing RNAi-res. TbPam16-HA (left) or ΔN-TbPam16-HA (right) in the background of RNAi targeting the wt TbPam16 (TbPam16-HA and ΔN-TbPam16-HA exclusive expressors). Insets: Immunoblot analysis of whole-cell extracts of uninduced (-) and 2 days induced (+) cells probed for wt TbPam16, RNAi-res. TbPam16-HA or ΔN-TbPam16-HA and EF1a as loading control. **(D)** Growth curve of uninduced (-Tet) and induced (+Tet) cells ectopically expressing RNAi-res. HA-TbPam16 in the background of RNAi targeting the wt TbPam16. Top inset: Immunoblot analysis of whole-cell extracts of uninduced (-) and 2 days induced (+) cells probed for myc-TbPam16 and EF1a as loading control. Bottom

inset: Northern blot of total RNA isolated from uninduced (-) and 2 days induced (+) cells probed for the mRNA of wt TbPam16. Ethidium bromide-stained rRNAs serve as loading control. Numerical data for panel (B) to (D) are available in S1 Data. MTS, mitochondrial targeting sequence; TMD, transmembrane domain.

length variant indicating that the lack of complementation is not due to insufficient expression of the protein (S6C Fig).

Alkaline carbonate extractions furthermore showed that as expected for an integral membrane protein, full-length TbPam16-HA is exclusively recovered in the pellet. In contrast, the major fraction of the truncated ΔN-TbPam16-HA is recovered in the supernatant indicating it is soluble.

In summary, the results in Fig 5A–5C suggest that the integration of TbPam18 and TbPam16 into the IM is essential for their function and thus link maxicircle replication to the IM.

### The IMS-exposed N-terminus of TbPam16 is functionally relevant

To find out, whether the very N-terminus of the 33 aa IMS-exposed domain is required for TbPam16 function, we expressed an N-terminally myc-tagged version of RNAi-resistant TbPam16 in the background of the TbPam16-RNAi cell line (Fig 5A). As in the case of the ΔN-TbPam16-HA variant, the myc-TbPam16 was not able to complement growth (Fig 5D). The myc-TbPam16 was highly expressed and a small amount was recovered in the crude mitochondrial fraction in a digitonin extraction. Moreover, essentially all myc-TbPam16 in this fraction was recovered in the pellet after an alkaline carbonate extraction as would be expected for an integral membrane protein (S6B Fig). These results suggest that the N-terminal myc-tag interferes with TbPam16 function indicating that at least part the IMS-exposed domain of TbPam16 is essential for its function.

### The J-domain of yeast Pam18 cannot complement the loss of the TbPam18 J-like domain

Classic Pam18 homologues involved in mitochondrial protein import contain the highly conserved tripeptide HPD in their J-domain (S7A Fig) [57], which is essential for the stimulation of the ATPase activity of their Hsp70 partners [62,63]. In contrast, TbPam18 has a degenerate J-domain containing the tripeptide HSD making it a J-like protein [63], a feature that is highly conserved within kinetoplastids (S7B Fig). Thus, we wondered (i) whether the HSD motif of TbPam18 is important for its function; and (ii) if the J-domain of yeast Pam18 (ScPam18) can take over the function of the J-like domain of TbPam18.

To that end, we generated a cell line allowing the exclusive expression of a TbPam18 variant in which the tripeptide HSD was mutated to HPD (TbPam18-S98P, Fig 6A). Intriguingly, this variant can fully complement the growth retardation caused by the RNAi-mediated depletion of TbPam18 (Fig 6B). Thus, TbPam18 can function with both, J or J-like domains.

To find out whether the intact J-domain of ScPam18 can replace the J-like domain of TbPam18, we used a chimeric protein consisting of the TMD of TbPam18 and the J-domain of ScPam18 (Tb/ScPam18) (Fig 6A). Expression of Tb/ScPam18 in TbPam18 RNAi background delayed the onset of the growth phenotype by 1 day but could not rescue the growth retardation at later time points (Fig 6C).

Because tagged versions of TbPam18 are not functional (S6A Fig), the variants tested above were untagged. Nevertheless, we analyzed the localization of N-terminally myc-tagged TbPam18 and Tb/ScPam18, which showed that about half of each variant is recovered in the

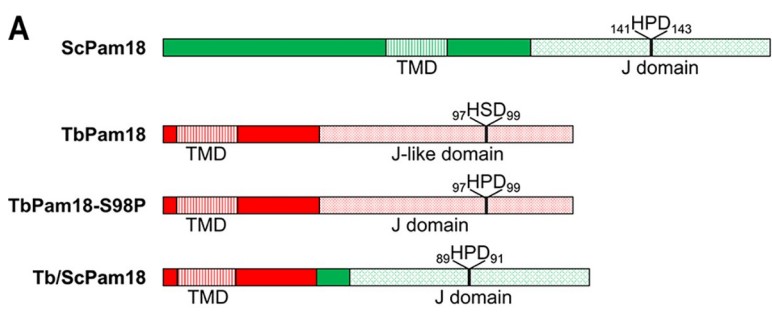

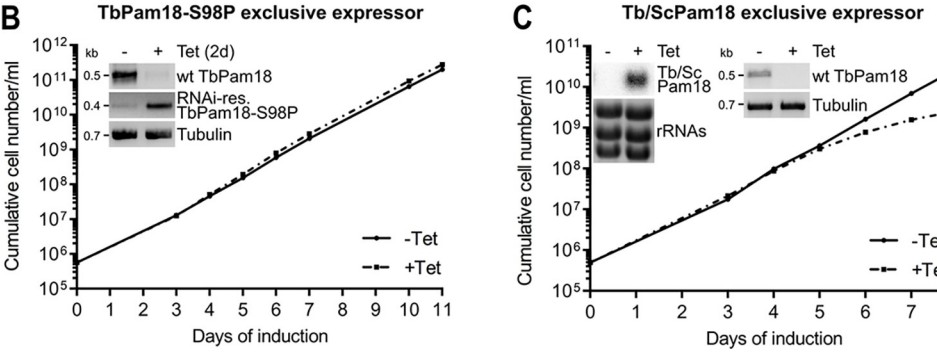

**Fig 6. The J-domain of ScPam18 cannot complement the loss of the J-like domain of TbPam18. (A)** Schematic representations of yeast (Sc) Pam18, TbPam18, and a mutated TbPam18 version, in which the J-like domain was altered to a J-domain by changing the serine residue at position 98 to a proline residue (TbPam18-S98P). Finally, a Tb/Sc fusion Pam18 (Tb/ScPam18), in which the J-like domain of TbPam18 was replaced by the J-domain of ScPam18 is shown. **(B)** Growth curve of uninduced (-Tet) and induced (+Tet) cells ectopically expressing RNAi-res. TbPam18-S98P in the background of RNAi targeting the endogenous wt TbPam18 (TbPam18-S98P exclusive expressor). Inset: RT-PCR products of wt TbPam18 and TbPam18-S98P mRNAs in uninduced (-) or 2 days induced (+) cells. Tubulin mRNA serves as loading control. **(C)** Growth curve of uninduced (-Tet) and induced (+Tet) cells ectopically expressing RNAi-resistant (RNAi-res.) Tb/ScPam18 in the background of RNAi targeting the endogenous wild-type (wt) TbPam18 (Tb/ScPam18 exclusive expressor). Inset on the left: Northern blot of total RNA extracted from uninduced (-) and 2 days induced (+) cells, probed for Tb/ScPam18, to confirm inducible ectopic expression. Inset on the right: RT-PCR product of the wt TbPam18 mRNA in uninduced (-) or 2 days induced (+) cells. Tubulin mRNA serves as loading control. Numerical data for panel (B) and (C) are available in S1 Data. TMD, transmembrane domain.

mitochondria-enriched fraction of a digitonin extraction (S7C Fig). Moreover, in an alkaline carbonate extraction, both TbPam18 versions present in the mitochondria-enriched fraction are exclusively recovered in the pellet indicating they are integrated into the IM. This strongly suggest that also untagged TbPam18 and Tb/ScPam18 are correctly localized.

In summary, these results show that some feature of the ScPam18 J-domain, other than the HSD or HPD tripeptide, is incompatible with the function of TbPam18.

## TbPam16 interacts with TbPam18 and 2 additional essential proteins

We have previously used TbPam18-HA and TbPam16-HA for SILAC co-immunoprecipitations experiments (CoIP) [26]. However, tagged TbPam18 is not functional (S6A Fig). Moreover, in the TbPam16-HA CoIP experiment, the only interactor identified was TbPam18 [26]. Thus, in hindsight, these experiments are difficult to interpret. Therefore, we repeated the TbPam16 SILAC CoIP with 2 modifications. First, we used the newly generated cell line allowing exclusive expression of functional TbPam16-HA (Fig 5C), and second, we analyzed both premix as well as postmix samples. Premix conditions means that differentially labeled

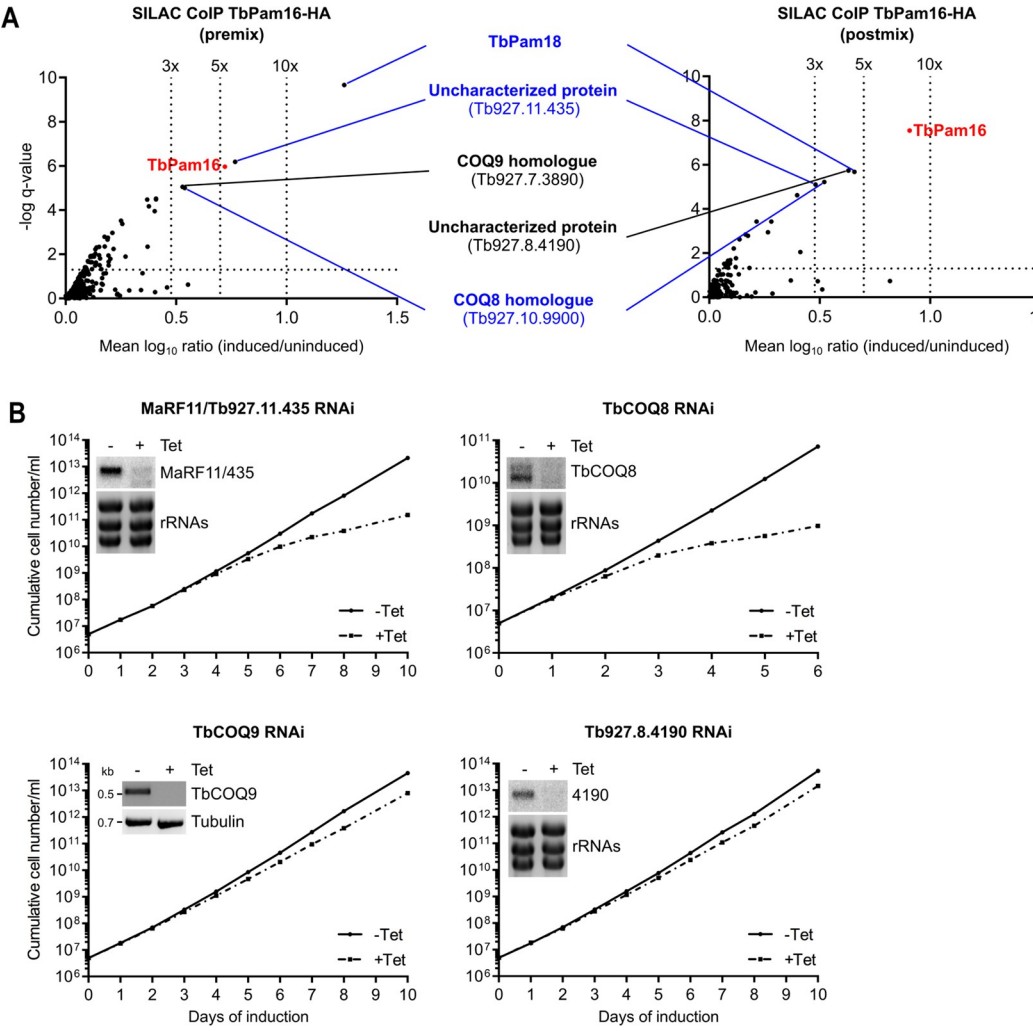

**Fig 7. TbPam16 interacts with TbPam18 and 2 other essential proteins. (A)** Volcano plots depicting proteins detected in SILAC-based quantitative MS analysis of TbPam16-HA CoIPs. In the experiment on the left, differentially labeled uninduced and induced cells were mixed and the resulting mixture was subjected to CoIP (premix). In the experiment on the right, CoIPs with uninduced and induced cells were done separately and the resulting eluates were mixed afterwards (postmix). The vertical dotted lines in the volcano plots indicate the specified enrichment factors. The horizontal dotted line indicates a rank-sum test significance level of 0.05. The bait TbPam16 is highlighted in red. Proteins that were significantly detected and enriched more than 3-fold in the pre- as well as the postmix experiments are labeled in blue. Proteins enriched more than 3-fold in either the pre- or the postmix experiment are labeled in black. **(B)** Growth curves of uninduced (-Tet) and induced (+Tet) MaRF11/Tb927.11.435, TbCOQ8, TbCOQ9, and Tb927.8.4190 RNAi cells. Insets in the MaRF11/Tb927.11.435, TbCOQ8, and Tb927.8.4190 RNAi growth curves: Northern blots of total RNA extracted from uninduced (-) and 2 days induced cells (+), probed for the respective mRNAs. Ethidium bromide-stained ribosomal RNAs (rRNAs) serve as loading controls. Inset in the TbCOQ9 RNAi growth curve: RT-PCR product of the TbCOQ9 mRNA in uninduced (-) or 2 days induced (+) cells. Tubulin mRNA serves as loading control. Numerical data for panel (A) and (B) are available in S2 Table and S1 Data, respectively. MS, mass spectrometry.

uninduced and induced cells are mixed prior to the CoIP, which preferentially detects stable interaction partners. In contrast, in the postmix sample, the eluates from separately generated, differentially labeled CoIPs are mixed, allowing the detection of both stable and more transient interaction partners [64].

In the premix experiment, TbPam16 was enriched 5.3-fold and in the postmix experiment 8.1-fold demonstrating that both CoIPs were successful (Fig 7A). Importantly, apart from the

bait TbPam16, the most enriched protein in both experiments was TbPam18 confirming the interaction between the 2 proteins.

Interestingly, in addition to TbPam18, 2 other proteins were enriched more than 3-fold in both experiments (Fig 7A). The first one is a trypanosome-specific, uncharacterized protein (Tb927.11.435) with a molecular weight of only 10.6 kDa and a remarkably high isoelectric point (pI = 10.6). This is interesting because a high pI is a characteristic often found in DNA-binding proteins. An AlphaFold structure prediction of Tb927.11.435 reveals that it consists of 4 consecutive in part amphiphilic α-helices (S8 Fig) [65]. The second protein, Tb927.10.9900 (Fig 7A), is homologous to yeast and human COQ8 and thus was termed TbCOQ8. COQ8 is a subunit of the coenzyme Q biosynthetic complex (complex Q), located on the matrix face of the IM [66]. While COQ8 appears to be required for coenzyme Q synthesis, its exact role is not known. Sequence homology suggests COQ8 might be a protein kinase; however, it likely does not function as a canonical kinase but may use its ATPase activity to stabilize complex Q [66]. Intriguingly, in the TbPam16-HA premix SILAC CoIP, Tb927.7.3890, a protein orthologous to complex Q subunit 9 (COQ9) was detected. COQ9 is a lipid-binding protein associated with COQ7 [66]. The presence of TbCOQ8 and TbCOQ9 among the most enriched proteins in the TbPam16-HA SILAC CoIPs is striking. However, only little is known about ubiquinone biosynthesis in *T. brucei* and neither of the 2 subunits have been previously analyzed. In the postmix experiment, the fifth protein enriched more than 3-fold was Tb927.8.4190, another trypanosome-specific, uncharacterized protein.

To further study the 4 newly identified TbPam16 interaction partners, we generated PCF RNAi cell lines. Growth curves revealed that knockdowns of Tb927.11.435 and TbCOQ8 cause a retardation in growth starting after 5 and 2 days of induction, respectively (Fig 7B). In contrast, RNAi against TbCOQ9 and Tb927.8.4190 only marginally affected growth. Hence, we focused on Tb927.11.435 and TbCOQ8 in further experiments.

## Tb927.11.435 ablation phenocopies the depletions of TbPam18 and TbPam16

Next, we investigated the fate of the kDNA upon Tb927.11.435 or TbCOQ8 depletion using the methods that were applied for TbPam16 and TbPam18 (Figs 2A, 3, and S5).

Quantification of DAPI-stained RNAi cells using fluorescence microscopy showed that in the Tb927.11.435 RNAi cell line the kDNA size was significantly reduced to 80% and 36% after 3 (prior to the onset of the growth retardation at day 4) to 6 days of RNAi induction, respectively (Fig 8A). On the contrary, ablation of TbCOQ8 was not found to significantly change the size of the kDNA over 4 days of RNAi induction. The kDNA network in the Tb927.11.435 RNAi cell line was further analyzed by quantitative PCR, which showed a decrease in the amount of maxicircles to 31% and less than 10% after 3 to 4 days of RNAi induction, respectively. In contrast, there was no significant change in the amount of minicircles for at least 6 days (Fig 8B). However, while the total amount of minicircles remained constant, they were progressively and essentially completely released from the kDNA network within 5 days of induction (Fig 8C).

Finally, we analyzed the Tb927.11.435 RNAi cell line using SILAC-based quantitative MS (S9 and S10 Figs), as was done for TbPam18 and TbPam16 RNAi cell lines (Figs 1B, S2, and S3). A total of 904 mitochondrial proteins [1,29,48] were detected. The most affected proteins were MRPs and OXPHOS subunits, of which 58% and 17% were depleted more than 1.5-fold, making up for 60% of all 1.5-fold depleted mitochondrial proteins (S9 Fig). For 7 protein categories consisting exclusively of nucleus-encoded proteins only 2 out of 205 proteins were more than 1.5-fold depleted (S10 Fig). These results are very similar to what has been observed in TbPam18 or TbPam16 depleted cells (Figs 1B, S2, and S3).

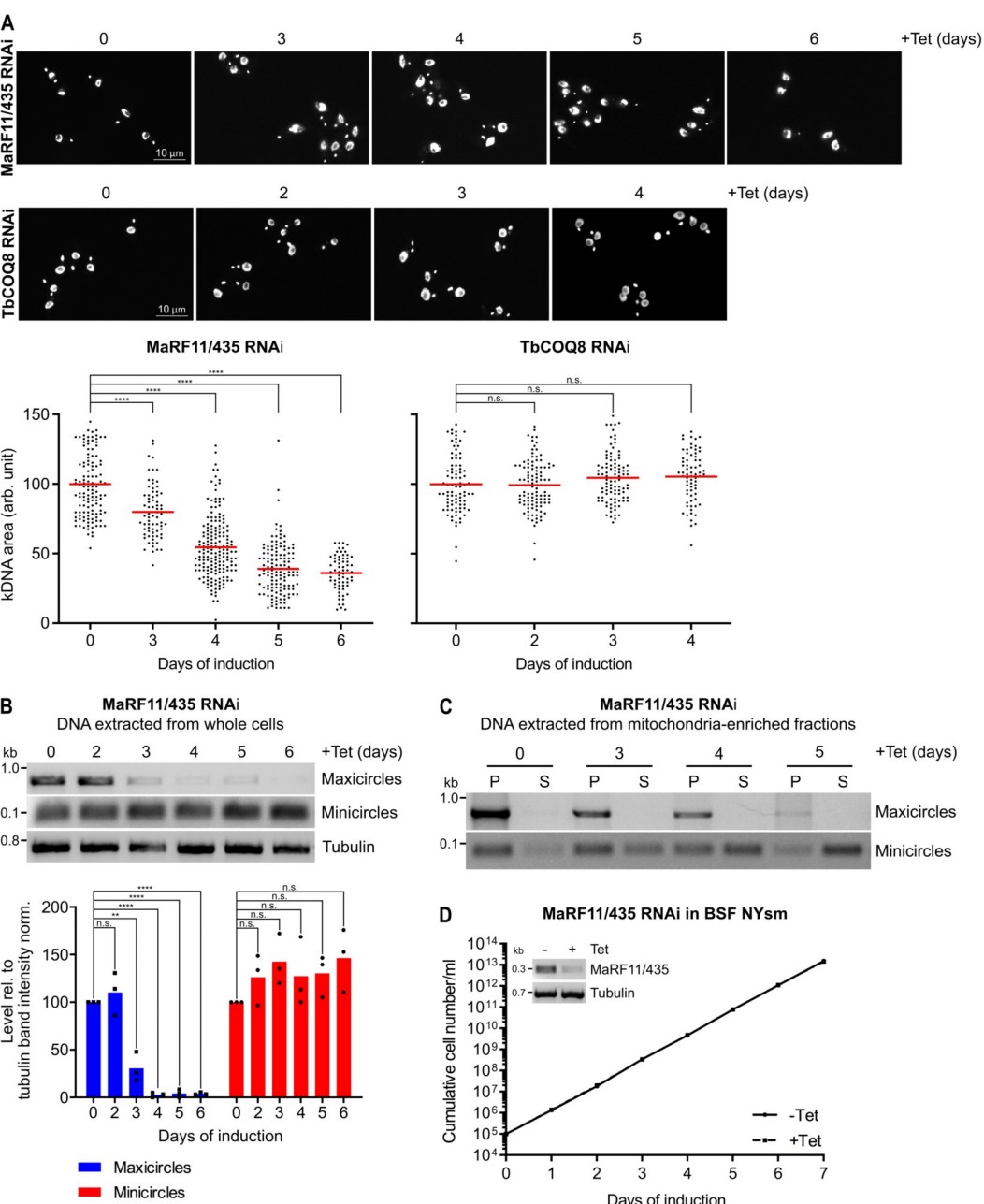

**Fig 8. Ablation of MaRF11/Tb927.11.435 phenocopies the depletions of TbPam18 and TbPam16 in PCF and BSF. (A)** Upper panels: Fluorescence microscopy analysis of DAPI-stained uninduced and 3 to 6 days induced MaRF11/ Tb927.11.435 (435) RNAi cells as well as uninduced and 2 to 4 days induced TbCOQ8 RNAi cells. Lower panels: Quantification of kDNA areas in 65 to 175 DAPI-stained RNAi cells. Red line indicates the mean of the kDNA size at each time point. The mean of the uninduced cells was set to 100%. n.s. = not significant, ****: *P* value <0.0001, as calculated by an unpaired two-tailed *t* test. **(B)** Upper panel: A quantitative PCR-based (qPCR) method was used to detect changes in steady-state levels of total mini- and maxicircles. Total DNA was extracted from uninduced and 2 to 6 days induced MaRF11/435 RNAi cells. This DNA was used as the template in PCRs amplifying specific mini- and maxicircle regions or the intergenic region of tubulin. PCR products were analyzed on agarose gels. Lower panels: Densitometric quantification of mini- and maxicircle abundance as detected by qPCR. The ratio of the mini- or maxicircle band intensity and the respective loading control (tubulin band intensity) was normalized (norm.) to the ratios of uninduced cells. Blue (maxicircles) and red (minicircles) bars represent the mean of 3 independent biological replicates. n.s.: not significant, **: *P* value <0.05, ****: *P* value <0.0001, as calculated by an unpaired two-tailed *t* test. **(C)** A quantitative PCR-based method used to analyze steady-state levels of kDNA-bound or free mini- and maxicircles. A digitonin-extracted, mitochondria-enriched pellet from uninduced and 3 to 5 days induced MaRF11/435 RNAi cells was solubilized in 1% digitonin. A subsequent centrifugation

step resulted in a pellet fraction (P) containing intact kDNA networks and a soluble fraction (S) containing free minicircles. DNA extracted from both fractions was used as template for PCR reactions amplifying specific mini- or maxicircle regions. PCR products were analyzed on agarose gels. **(D)** Growth curve of uninduced (-Tet) and induced (+Tet) BSF NYsm RNAi cell line ablating MaRF11/435. Inset: RT-PCR product of MaRF11/435 mRNA in uninduced (-) or 2 days induced (+) cells. Tubulin mRNA serves as loading control. Numerical data for panel (A), (B), and (D) are available in S1 Data. BSF, bloodstream form; kDNA, kinetoplast DNA; NYsm, New York single marker; PCF, procyclic form.

To quantify the similarities the depletion of TbPam18, TbPam16, and Tb927.11.435 caused on the mitochondrial proteome, we performed a correlation analysis comparing all 3 data sets. Moreover, we included a previously published SILAC RNAi data set for TbTim17, the core component of the single trypanosomal TIM complex (S11 Fig) [22]. This allowed to compare how the mitochondrial proteome reacted to depletion of a general IM import factor versus factors involved in maxicircle replication. The results showed that the Spearman's rank correlation factors among the TbPam18, TbPam16, and Tb927.11.435 data sets were between 0.69 and 0.73 (S11 Fig), which is very high considering the maximal possible correlation is 1.0. In contrast, the correlation between the TbTim17 data set and the 3 other data sets was much lower ranging from 0.21 to 0.3 (S11 Fig).

These results show that depletion of TbPam18, TbPam16, and Tb927.11.435 caused the same proteomic phenotypes, strongly suggesting that the 3 proteins are involved in the same biological process. Moreover, the results also confirm that depletion of TbTim17, a general IM import factor, results in different changes to the mitochondrial proteome than were observed for the 3 other factors.

Thus, depletion of Tb927.11.435 exactly phenocopied the results observed in induced TbPam16 and TbPam18 PCF RNAi cells, which is why we named it maxicircle replication factor of 11 kDa (MaRF11) (Figs 2, 3, and 8). Moreover, RNAi-induced knockdown of MaRF11 in BSF NYsm cells does not cause a change in the growth rate, which is identical to what is observed in BSF TbPam18 RNAi or TbPam16 dKO cells (Figs 4 and 8D).

### TbHslV controls the level of MaRF11

One way in which TbPam18 and TbPam16 could regulate the activity of MaRF11 would be by controlling its degradation, reminiscent of TbPIF2, the levels of which are controlled by the mitochondrial protease TbHsIVU [47]. Indeed, RNAi-mediated ablation of the TbHslV subunit of TbHslVU results in a growth retardation (Fig 9A) and a concomitant accumulation of giant kDNAs as observed previously (Fig 9B) [47]. Under the same conditions, analogous to TbPIF2, a continuous increase in the levels of MaRF11 for up to 6 days after induction was observed, whereas the levels of TbPam16 and EF1a remained constant (Fig 9C).

### Discussion

Our study has shown that TbPam18 and TbPam16 are essential for replication of the maxicircle component of the kDNA (Fig 2). This observation closes an important gap in the previously proposed evolutionary scenario explaining why trypanosomes have a single bifunctional TIM complex only [26]. Our finding is unexpected, because neither Pam18 nor Pam16 orthologues have ever been associated with the biogenesis of the mitochondrial genome before. Thus, while phylogenomics classifies TbPam18 and TbPam16 as bona fide Pam18 and Pam16 orthologues [26], the 2 proteins have switched their function during evolution, from protein import to mitochondrial genome replication. It is worth noting that this change in function could only be discovered experimentally and was not predicted in silico. Interestingly, yeast Pam18 has also been associated with another function than mitochondrial protein import. It

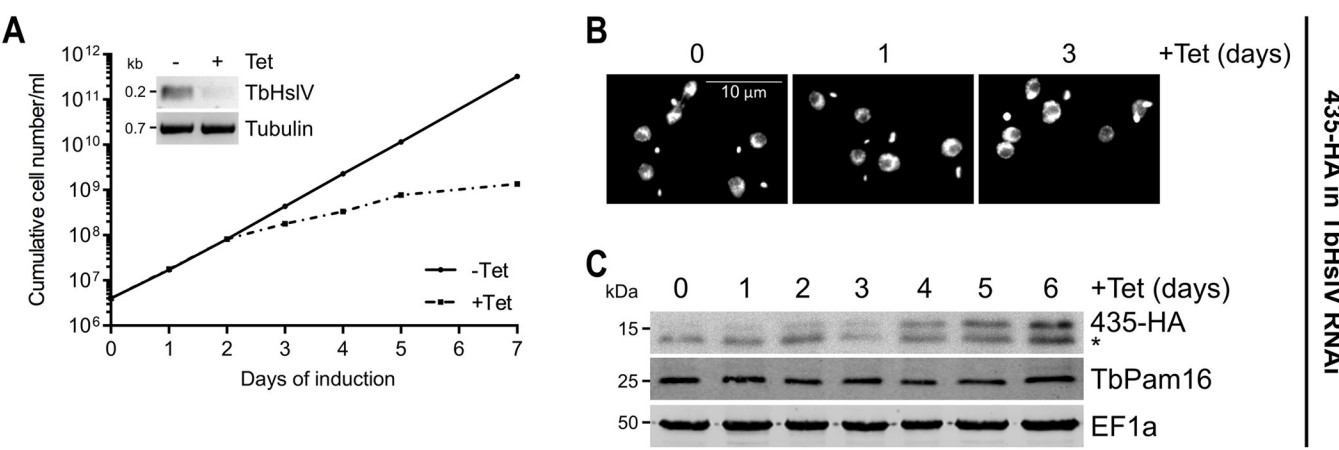

**Fig 9. TbHslV controls the level of MaRF11/Tb927.11.435. (A)** Growth curve of uninduced (-Tet) and induced (+Tet) cells expressing MaRF11/435-HA in the background of TbHslV RNAi. Inset shows the RT-PCR product of the TbHslV mRNA in uninduced (-) or 2 days induced (+) cells. Tubulin mRNA serves as loading control. **(B)** Fluorescence microscopy analysis of the DAPI-stained uninduced and 1 or 3 days induced cell line explained in (A). **(C)** Immunoblot analysis of steady-state protein levels of MaRF11/435-HA and TbPam16 in TbHslV RNAi background over 6 days of induction. EF1a serves as loading control. Asterisk indicates an unspecific band. Numerical data for panel (A) are available in S1 Data.

can stimulate the mHsp70-dependent assembly of respiratory super-complexes, when present as a homodimer [67]. However, in contrast to TbPam18, yeast Pam18 is still an essential component of the PAM and thus has a dual function.

Among all proteins known to be involved in kDNA maintenance and replication, depletion of only 4 preferentially affects the maxicircles: the mitochondrial RNA polymerase (mtRNAP) [68], the mitochondrial DNA primase 1 (TbPRI1) [52], the mitochondrial DNA helicase (TbPIF2) [47], and the mitochondrial heat shock proteins TbmHsp70/TbmHsp40 [53]. We have now discovered 3 additional factors: TbPam18, TbPam16, and the TbPam16-interacting MaRF11. Each of them is essential for normal growth of PCF *T. brucei* [26] (Fig 7B) and their depletion specifically affects replication of the maxicircles prior to the onset of the growth arrest (Figs 2B and 8B). Interestingly, similar to what has been observed upon ablation of the maxicircle replication factors mentioned above, total minicircle levels remain constant after the loss of maxicircles upon TbPam18, TbPam16, and MaRF11 depletion (Figs 2B and 8B). Moreover, depletion of TbPRI1 [52] and TbmHsp70/TbmHsp40 [53], but not of TbPIF2 [47], causes a rapid shrinkage of kDNA disks. The same was observed after TbPam18, TbPam16, and MaRF11 ablation (Figs 2A and 8A). A reduction in maxicircles alone, which make up only 10% of the kDNA network, cannot explain this shrinkage. In fact, while maxicircles are selectively depleted in the TbPIF2 RNAi cell line, its kDNA network remains intact [47]. It has been suggested for TbPRI1 [52] and TbmHsp70/TbmHsp40 [53] depletion that the shrinkage of the kDNA network is due to the detachment of minicircles from the kDNA disk. The free minicircles are then replicated but cannot reattach to the maxicircle-depleted kDNA network. As a consequence, the kDNA network shrinks and free minicircles accumulate in the mitochondrial matrix [52,53], which is exactly what is also observed in the TbPam18, TbPam16, and the MaRF11 RNAi cell lines (Figs 3 and 8C).

How can we explain that TbPam16, TbPam18, and MaRF11 are dispensable in the BSF of *T. brucei*? The 2 life cycle stages show many differences, including optimal growth temperatures (27°C for PCF/37°C for BSF) and generation times (10 to 12 h for PCF/5 to 6 h for BSF). Moreover, the PCF can undergo cytokinesis without completion of mitosis, whereas in the BSF a mitotic block inhibits cytokinesis but not kDNA replication [69]. Thus, it would not be

surprising if these differences may require some life cycle stage-specific adaptations in the regulation of kDNA replication.

TbPam18 and TbPam16 are integral IM proteins, each containing a single TMD [26] that is essential for their function (Fig 5). This contrasts with yeast, where the TMD of Pam18 is dispensable and where Pam16 does not even have a TMD [70]. Our results functionally connect maxicircle replication to the IM. This is unexpected as maxicircles are replicated while remaining interlocked with the kDNA network and proteins involved in their replication generally localize to the kDNA disk [71]. The only known integral IM protein associated with kDNA inheritance is p166, a subunit of the TAC [72,73]. The TAC is essential for kDNA segregation but not for its replication [37,74]. Knockdown of TAC subunits leads to enlarged, overreplicated kDNAs in a few cells [37,74], which is not what is observed upon TbPam16, TbPam18, and MaRF11 depletion. It is therefore unlikely that these 2 proteins are involved in kDNA segregation and their IM localization must be explained in a different way.

J-domain family proteins are known regulators of a plethora of biological processes by selecting client proteins for Hsp70-type chaperones [62,75,76]. Interestingly, *T. brucei* has a greatly expanded mitochondrial J-domain protein family consisting of at least 38 members [1,77–81]. One reason for this could be that they may be required for functional differentiation of the single TbmHsp70.

TbPam16, just as its yeast counterpart, is a J-like protein. However, while Pam18 in yeast contains an intact J-domain with a conserved HPD motif [70], TbPam18 is a J-like protein containing an HSD tripeptide. Expression of a chimeric protein, in which the J-like domain of endogenous TbPam18 was replaced by the J-domain of yeast Pam18, failed to restore growth in a TbPam18 RNAi cell line (Fig 6A and 6C). Surprisingly, the change of TbPam18's HSD to an HPD does not affect the functionality of the protein (Fig 6A and 6B). Thus, the divergence that prevents the functional interchangeability of the TbPam18 J-like domain with the J-domain of yeast Pam18 must have occurred outside the tripeptide motif.

Since the HPD motif is essential for the interaction of J-domains with their Hsp70 partners [82], the J-like domains of TbPam18 and TbPam16 likely cannot stimulate the ATPase activity of TbmHsp70, at least not directly. Thus, the change in function of TbPam18 during evolution must have allowed the inactivation of the conserved tripeptide from HPD to HSD indicating that TbPam18 and TbPam16 function independently of mHsp70. This is reminiscent to what has been suggested for *Arabidopsis thaliana*, which has 21 J-like proteins involved in various processes, most of which function independently of Hsp70 [83,84]. In line with this, pulldown experiments of TbPam16 prominently recover TbPam18, but neither TbmHsp70 nor other J-domain protein family members (Fig 7A). Instead, the SILAC CoIPs recovered the 2 essential proteins MaRF11 and TbCOQ8 indicating they might be client proteins (Fig 7A). This implies that the TbPam18/TbPam16 heterodimer binds to a few clients only. Ablation of TbCOQ8 inhibits growth of PCF trypanosomes, but does not affect the size of the kDNA (Figs 7B and 8A). However, ablation of the soluble, basic protein MaRF11 phenocopies what is observed in TbPam18 and TbPam16 RNAi cell lines (Figs 7B, 8A, and 8B). We have also shown that MaRF11 is a substrate of the mitochondrial proteasome orthologue TbHslVU, since depletion of the TbHslVU subunit TbHslV increases MaRF11 levels (Fig 9C). Thus, both MaRF11 and the previously characterized TbPIF2, the only 2 TbHsIVU substrates known to date, are essential maxicircle replication factors explaining the three to 4-fold accumulation of maxicircles in the absence of TbHslV [28]. However, the TbHsIVU substrate(s) causing the previously reported 20-fold accumulation of minicircles in the absence of TbHslVU remain unknown [28]. In summary, the identification of MaRF11 as a second substrate of TbHslVU underscores the central role the mitochondrial proteasome plays in controlling kDNA replication.

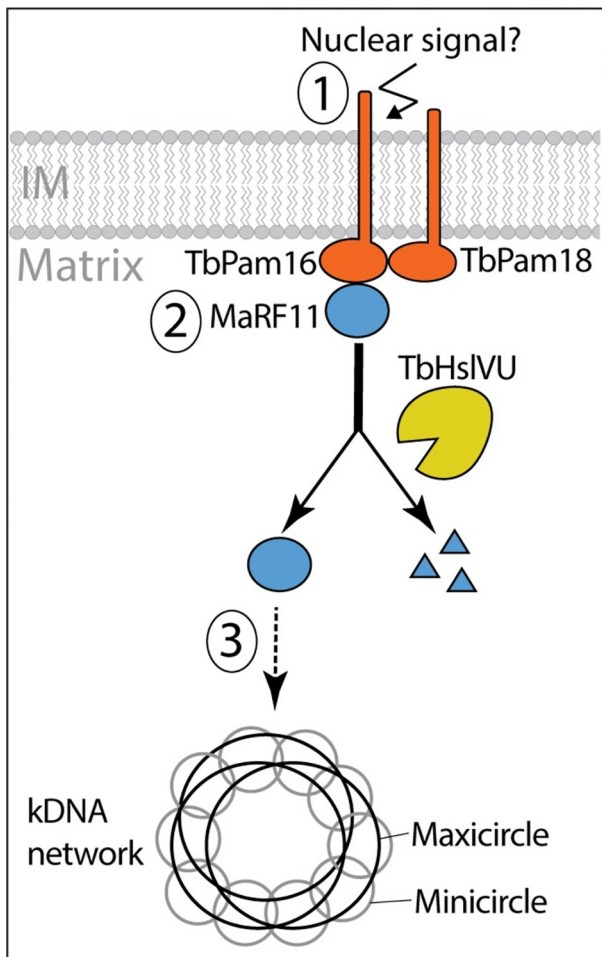

**Fig 10. Working model of maxicircle replication control by MaRF11.** The integral membrane proteins TbPam18 and TbPam16, together with the soluble matrix protein MaRF11, are part of membrane-bound complex required for maxicircle replication. MaRF11 acts downstream of TbPam18 and TbPam16. The levels of MaRF11 are regulated by proteolytic degradation via TbHslVU. Outstanding questions are highlighted. These include the identity of a putative nuclear signal (1), how such a signal may regulate the binding of MaRF11 to TbPam18 and TbPam16 (2) and whether MaRF11 directly mediates maxicircle replication or whether it activates other as yet unknow replication factors (3). kDNA, kinetoplast DNA; IM, inner membrane.

Based on the results described above, we propose the following working model for a mitochondrial inner membrane-bound protein complex involved in the replication of the maxicircle portion of the kDNA (Fig 10). The complex includes the soluble protein MaRF11 and the integral IM proteins TbPam18 and TbPam16. The latter 2 probably form a heterodimer and have their J-like domains exposed to the matrix. All 3 proteins are individually essential for maxicircle replication in procyclic trypanosomes. The fact that MaRF11 is a soluble matrix protein suggests that it may function downstream of the integral IM proteins TbPam18 and TbPam16. Furthermore, we have shown that MaRF11 levels are controlled by proteolytic degradation by TbHslVU.

In *T. brucei*, the mitochondrial S-phase is precisely coordinated with the nuclear S-phase by an as yet unknown signal [85]. Since not only the TMDs of TbPam18 and TbPam16 but also the IMS-exposed N-terminus of TbPam16 are essential for normal growth, it is tempting to speculate that the TbPam18/TbPam16/MaRF11 complex may play a role in this coordination.

However, some aspects of the model remain speculative. For example, how MaRF11 controls maxicircle replication is still unknown. Its high pI would be consistent with a direct binding to the kDNA, but it is also possible that MaRF11 is only one component of a more elaborate maxicircle replication pathway that involves other factors further downstream. However, pulldown experiments aimed at identifying such factors in a cell line exclusively expressing tagged MaRF11 were not successful. Spatial proteomic analyses of the procyclic and bloodstream forms of *T. brucei* showed that at steady-state TbPam18, TbPam16, and MaRF11 co-fractionate with IM marker proteins, consistent with the TbPam16 pulldown experiment (Fig 7A) [86]. It would support our model if the 3 proteins were concentrated in the IM around the kDNA region. Unfortunately, despite the use of signal enhancement protocols, we were unable to detect any of the 3 tagged proteins by immunofluorescence microscopy. This suggests that, consistent with our inability to detect any of the 3 proteins by MS in our global SILAC proteomics studies, they are of too low abundance to be detected by immunofluorescence microscopy. Despite these limitations the proposed model can serve as a guide for future experiments aimed at providing more insights into the intricate process of how maxicircles and the kDNA in general are replicated.

## Materials and methods

### Transgenic cell lines

Transgenic *T. brucei* cell lines are either based on the PCF strain 29–13 or the BSF strain NYsm [60]. PCF cells were grown in SDM-79 [87] supplemented with 10% (v/v) fetal calf serum (FCS) at 27°C. BSF cells were cultivated in HMI-9 [88] containing 10% (v/v) FCS at 37°C.

RNAi against TbPam18 (Tb927.8.6310) and TbPam16 (Tb927.9.13530) has been described previously [26]. For complementation experiments, synthetic genes (Biomatik) were used. The codons in regions of the open reading frame (ORF) that are targeted by RNAi were changed such that their transcripts are RNAi resistant (RNAi-res.) but still translate into the same amino acid sequence as in the endogenous proteins. To produce constructs allowing expression of N-terminally truncated TbPam18 (ΔN-TbPam18) and TbPam16 (ΔN-TbPam16), the corresponding DNA fragments were amplified from the synthetic genes. To ensure targeting to mitochondria, the MTS of trypanosomal mitochondrial Hsp60 (TbmHsp60, Tb927.10.6510) was cloned in front of the truncated constructs. For the RNAi-res. Tb/ScPam18 fusion protein and RNAi-res. TbPam18-S98P, additional synthetic genes (Biomatik) were used. To generate RNAi-res. Tb/ScPam18, the first 138 nucleotides of RNAi-res. TbPam18 were fused to the last 213 nucleotides of wild-type yeast Pam18 (YLR008C). To generate TbPam18-S98P, the cytosine at position 292 of the nucleotide sequence of RNAi-res. TbPam18 was exchanged against a thymine. Sequences of all synthetic genes are shown in S12 Fig.

To generate plasmids for ectopic expression of untagged, N- or C-terminally triple c-myc- or HA-tagged, RNAi-res., full-length or N-terminally truncated TbPam18 or TbPam16 variants, Tb/ScPam18, TbPam18-S98P, MaRF11 and TbTim54 (Tb927.6.2470) the complete or truncated ORFs of the respective genes were amplified by PCR. The PCR products subsequently were cloned into a modified pLew100 vector [60,89], which contains a puromycin resistance cassette and either no epitope tag or triple c-myc- or HA-tags [90].

MaRF11, TbCOQ8, TbCOQ9, Tb927.8.4190, and TbHslV (Tb927.11.10240) RNAi cell lines were generated using the same pLew100-derived vector described above. This vector allows the generation of a stem-loop construct by the insertion of the RNAi target regions in opposite directions and a 460 nucleotide (nt) spacer fragment forming the loop. The RNAi targets the

indicated nts of the ORFs of MaRF11 (Tb927.11.435, nt 14–255) and TbCOQ8 (nt 353–536) or the 3′ untranslated regions (UTRs) of TbCOQ9 (nt (+49)—(+543)), Tb927.8.4190 (nt (+253) —(+458)), and TbHslV (nt (+114)—(+609)). To ensure efficient transcription of the RNAi construct in the NYsm BSF strain, the procyclin promotor in the above described MaRF11 RNAi plasmid was exchanged against the rRNA promotor.

To generate the TbTim54 conditional dKO cell line, 90 nts up- and downstream of the TbTim54 alleles were fused to either a hygromycin (hygro) or a neomycin (neo) resistance cassette. The first TbTim54 allele was replaced by hygro resulting in the single knockout (sKO). To generate the dKO, the second TbTim54 allele was replaced by neo. In a third cloning step, this TbTim54 dKO cell line was transfected with the plasmid for ectopic expression of TbTim54-HA described above.

The NYsm BSF cell line containing the TbPam16 dKO was generated by fusing the 500 nts up- and downstream of the TbPam16 alleles to the N- or C-terminus of hygro or the blasticidin (blast) resistance cassette, respectively. The first TbPam16 allele was replaced by hygro and the second TbPam16 allele was replaced by blast.

## Antibodies

Polyclonal rabbit antiserum against TbPam16 was commercially produced (Eurogentec, Belgium) using amino acids 153–167 (VKDSHGNSRGNDAMW) as antigen. For western blots (WB), the TbPam16 antiserum was used at a 1:500 dilution. Commercially available antibodies used in this study were: Mouse anti-c-myc (Invitrogen, dilution WB 1:2'000), mouse anti-HA (Sigma-Aldrich, dilution WB 1:5'000), and mouse anti-EF1a (Merck Millipore, dilution WB 1:10'000). Polyclonal rabbit anti-ATOM40 (dilution WB 1:10'000) and polyclonal rabbit anti-Cyt C (dilution WB 1:100) were previously produced in our laboratory [81,91]. Secondary antibodies used: Goat anti-mouse IRDye 680LT conjugated (LI-COR Biosciences, dilution WB 1:20'000) and goat anti-Rabbit IRDye 800CW conjugated (LI-COR Biosciences, dilution WB 1:20'000). For detection of 435-HA on immunoblots, HRP-coupled anti-mouse secondary antibodies (Sigma) were used.

## Digitonin extraction

Cell lines were induced with tetracycline for 1 day prior to the experiment to ensure expression of epitope-tagged proteins. To selectively solubilize the plasma membrane, $1 \times 10^8$ cells were incubated at 4˚C for 10 min in a buffer containing 0.6 M sorbitol, 10 mM Tris-HCl (pH 7.5), 1 mM EDTA (pH 8.0), and 0.015% (w/v) digitonin. A mitochondria-enriched pellet was separated from a supernatant that is enriched in cytosolic proteins by centrifugation (6'800 g, 5 min, 4˚C). Equivalents of $2 \times 10^6$ cells of each fraction were analyzed by SDS-PAGE and western blotting.

## Alkaline carbonate extraction

A digitonin-extracted, mitochondria-enriched pellet was resuspended in 100 mM $Na_2CO$ (pH 11.5) and incubated at 4˚C for 10 min. Centrifugation (100'000 g, 10 min, 4˚C) yielded in a pellet enriched in integral membrane proteins and a supernatant enriched in soluble or loosely membrane-associated proteins. Equivalents of $2 \times 10^6$ cells of each fraction were analyzed by SDS-PAGE and western blotting.

## Fluorescence microscopy and kDNA area quantification

TbPam18 and TbPam16 RNAi cells were fixed with 4% paraformaldehyde in PBS, postfixed in cold methanol, and mounted using VectaShield containing 4′,6-diamidino-2-phenylindole

(DAPI) (Vector Laboratories). Images were acquired by a DMI6000B microscope and a DFC360 FX monochrome camera (both Leica Microsystems).

Images were analyzed using ImageJ [92]. The kDNA size analysis was performed on binarized 8-bit format images. The size of particles was measured in arbitrary units (a.u.) and kDNA particles >0.0 a.u. and <0.75 a.u. were included in the analysis. Boomerang shaped, dividing kDNAs and randomly picked up particles were manually removed from the analysis. Significance of these results was calculated by an unpaired two-tailed *t* test.

### RNA extraction, RT-PCR, and northern blotting

Acid guanidinium thiocyanate-phenol-chloroform extraction to isolate total RNA from uninduced and 2 days induced RNAi cells was done as described elsewhere [93]. To determine RNAi efficiency, the extracted RNA was either utilized for RT-PCR or separated on a 1% agarose gel in MOPS buffer containing 0.5% formaldehyde for subsequent northern blotting. Northern probes were generated from gel-purified PCR products corresponding to the RNAi inserts or the overexpressed proteins described above, and radiolabeled by means of the Prime-a-Gene labeling system (Promega).

### DNA extraction, Southern blotting, and quantitative PCR

For DNA isolation, $5 \times 10^7$ cells were resuspended in NTE buffer (100 mM NaCl, 10 mM Tris (pH 7.5), and 5 mM EDTA) containing 0.5% SDS for cell lysis and 0.2 mg/ml RNase A to degrade RNA. After incubation for 1 h at 37˚C, 1 mg/ml proteinase K was added, followed by 2 h of incubation at 37˚C. DNA was isolated by phenol-chloroform extraction and subsequent ethanol precipitation.

For Southern blotting, 5 µg of DNA were digested overnight at 37˚C with HindIII and XbaI. Digested DNA was separated in a 1% agarose gel in 1× TAE buffer. Gel processing and blotting was done as described elsewhere [51,94]. For kDNA detection, sequence-specific mini- and maxicircle probes were generated by PCR. The minicircle probe was a 0.1 kb stretch of the conserved minicircle sequence [94]. A 1.4 kb fragment served as the maxicircle probe [51,95]. For normalization, a tubulin probe binding to a 3.6 kb stretch within the intergenic region between α- and β-tubulin was used [94]. Probes were radiolabeled by means of the Prime-a-Gene labeling system (Promega).

To determine total mini- and maxicircles or free minicircle levels by quantitative PCR, DNA was either isolated from whole cells or from fractionated digitonin-extracted, mitochondria-enriched pellets and used as the template in a PCR utilizing the same primers as for the Southern blot probes.

### SILAC RNAi and SILAC CoIP experiments

TbPam18, TbPam16, and MaRF11 RNAi cells or cells exclusively expressing TbPam16-HA were washed in PBS and resuspended in SDM-80 [96] containing 5.55 mM glucose, 10% dialyzed FCS (BioConcept, Switzerland) and either light ($^{12}C_6/^{14}N_\chi$) or heavy ($^{13}C_6/^{15}N_\chi$) isotopes of arginine (1.1 mM) and lysine (0.4 mM) (Euroisotope). The cells were grown in SILAC medium for 6 to 10 doubling times to ensure a complete labeling of all proteins with heavy amino acids. For the SILAC RNAi and the premix SILAC CoIP, uninduced and induced (4 days for SILAC RNAi, 2 days for SILAC CoIP) cells were mixed in a one-to-one ratio. For the postmix SILAC CoIP, uninduced and induced cells were kept separately. From all samples, digitonin-extracted, mitochondria-enriched pellets were generated. For the SILAC RNAi experiments, these pellets were processed as described previously including tryptic in solution digestion [97] and then analyzed by liquid chromatography–mass spectrometry (LC–MS).

TbPam18 and TbPam16 SILAC RNAi experiments were done in 4 biological replicates including a label-switch.

For the SILAC CoIP experiments, mitochondria-enriched digitonin pellets were solubilized in a buffer containing 20 mM Tris-HCl (pH 7.4), 0.1 mM EDTA, 100 mM NaCl, 10% glycerol, 1X Protease Inhibitor mix (Roche, EDTA-free), and 1% (w/v) digitonin for 15 min at 4˚C. After centrifugation (21'000 g, 15 min, 4˚C), the lysate was transferred to HA bead slurry (anti-HA affinity matrix, Roche), which had been equilibrated in wash buffer (20 mM Tris-HCl (pH 7.4), 0.1 mM EDTA, 1 mM NaCl, 10% glycerol, 0.2% (w/v) digitonin). Incubation in an end-over-end shaker for 2 h at 4˚C was followed by removal of the supernatant containing the unbound proteins. After washing the bead slurry 3 times with wash buffer, the bound proteins were eluted by boiling the resin for 5 min in 2% SDS in 60 mM Tris-HCl (pH 6.8). In case of the postmix SILAC CoIP, eluates of uninduced and induced cells were now mixed in a one-to-one ratio. All eluates were further prepared for analysis by LC–MS as has been described in detail elsewhere [98]. TbPam16-HA pre- and postmix SILAC CoIP experiments were done in 4 biological replicates including label-switches.

## LC–MS and data analysis

LC–MS analyses of tryptic peptide mixtures from all experiments were performed using a Q Exactive Plus mass spectrometer connected to an UltiMate 3000 RSLCnano HPLC system (Thermo Fisher Scientific, Germany) as described before [26] with minor modifications. The software package MaxQuant [99,100] (version 1.6.3.4 for TbPam18 and TbPam16 SILAC RNAi, 2.0.2.0 for TbPam16 SILAC IP, and 2.5.1.0 for MaRF11 SILAC RNAi data) was used for protein identification and SILAC-based relative quantification. Mass spectrometric raw data were searched against a database containing the protein sequences for *T. brucei* TREU927 as provided by the TriTrypDB (https://tritrypdb.org; version 8.1 for TbPam18/TbPam16 SILAC RNAi and TbPam16 SILAC IP experiments, and version 55 for MaRF11 SILAC RNAi samples). Protein identification and quantification was based on ≥1 unique peptide and ≥1 ratio count, respectively. For all other parameters, MaxQuant default settings were used, including carbamidomethylation of cysteine as fixed modification, N-terminal acetylation and oxidation of methionine as variable modifications, and Lys8/Arg10 as heavy labels. The options "requantify" and "match between runs" were enabled.

MaxQuant result files were processed with python using pandas (version 1.5.3; https://pandas.pydata.org) as well as numpy (version 1.24.2; https://numpy.org/), seaborn (version 0.11.2; https://seaborn.pydata.org), scipy (version 1.10.0; https://www.scipy.org/), and matplotlib (version 3.6.3; https://matplotlib.org/) for data analysis and visualization. Data analysis of all experiments was based on protein abundance ratios calculated by MaxQuant.

To identify proteins affected by ablation of TbPam16 and TbPam18 following RNAi induction, MaxQuant protein ratios were first normalized replicate-wise by adjusting the summed ratios to the highest value, followed by cyclic loess normalization [101] of $log_2$-transformed protein ratios as implemented in the R Bioconductor (version 3.17) package "affy" [102] (version 1.78.2). Values missing in 1 or 2 out of 4 replicates were imputed using the DIMA package [103] (https://github.com/kreutz-lab/DIMAR). To identify proteins with significantly altered abundance upon RNAi induction, the "linear models for microarray data" (limma) approach [104,105] (version 3.28.14) was applied. *P* values were corrected for multiple testing following the Benjamini–Hochberg method [106]. Proteins affected by the RNAi-induced ablation of MaRF11 were identified based on normalized protein abundance ratios determined by MaxQuant using the limma approach and corrected *P* values as described above.

To identify proteins significantly enriched in TbPam16 pre- and postmix SILAC CoIP experiments, the rank sum method [107,108] as implemented in the R package "RankProd" [109] (version 3.24.0) was applied. The rank sum, defined as the arithmetic mean of the ranks of a protein in all replicates, was converted into FDR-controlled q-values.

For information about proteins identified and quantified, see S1–S3 Tables.

## Supporting information

**S1 Fig. Schematic depiction of mitochondrial inner membrane protein translocases and organization of the mitochondrial genome, termed kinetoplast DNA (kDNA) in T. brucei. (A)** Schematic depiction of the yeast TIM23 and TIM22 complexes (left) and the single trypanosomal TIM complex (right). In the trypanosomal TIM complex, the subunits specifically associated with presequence pathway are indicated in broken lines. Subunits of the presequence-associated import motor (PAM) are indicated in bold red lines. Unique and homologous subunits between the 2 species are indicated in gray and orange, respectively. The non-homologous J domain proteins Pam18/16 and TbPam27 are indicated in bold. (B) Organization of the single unit kDNA of *T. brucei*. The kDNA is a disk consisting of an intercalated network of maxi- and minicircles, which is physically connected, across the outer and the inner membrane, with the basal body of the flagellum via the tripartite attachment complex (TAC). Minicircles are replicated via theta structures after they have been released from the center of the network into the kinetoflagellar zone. After replication, they are reattached to the kDNA disk at the antipodal sites. Maxicircles in contrast are replicated while remaining attached to the kDNA disk.
(TIF)

**S2 Fig. TbPam18 RNAi does not affect protein subgroups exclusively consisting of nucleus-encoded proteins.** Mitochondria-enriched fractions of uninduced and 4 days induced TbPam18 RNAi cells were analyzed by SILAC-based quantitative mass spectrometry (same data set as in Fig 1B). **(A–F)** The indicated subgroups of mitochondrial proteins are highlighted in the indicated colors. The number of all more than 1.5-fold depleted proteins and the total number of all detected proteins for each subgroup are indicated in parentheses at the top of each panel. Numerical data for panel (A) to (F) are available in S1 Table.
(TIF)

**S3 Fig. TbPam16 RNAi does not affect protein subgroups exclusively consisting of nucleus-encoded proteins.** Mitochondria-enriched fractions of uninduced and 4 days induced TbPam16 RNAi cells were analyzed by SILAC-based quantitative mass spectrometry (same data set as in Fig 1B). **(A–F)** The indicated subgroups of mitochondrial proteins are highlighted in the indicated colors. The number of all more than 1.5-fold depleted proteins and the total number of all detected proteins for each subgroup are indicated in parentheses at the top of each panel. Numerical data for panel (A) and (F) are available in S1 Data.
(TIF)

**S4 Fig. TbTim54 is not required to maintain kDNA integrity. (A)** Growth curve of a TbTim54 conditional double knockout (cond. dKO) cell line. While both TbTim54 alleles are knocked out in uninduced (-Tet) as well as induced (+Tet) cells, TbTim54-HA is only ectopically expressed when tetracycline is present. Inset: Verification of single knockout (sKO) and dKO by PCR using a primer pair to amplify the TbTim54 ORF (approximately 2.0 kilobases (kb)), the hygromycin (hygro, approximately 1.5 kb), or the neomycin (neo, approximately 1.3 kb) resistance cassettes at the same time. Hygro was used to replace the first and neo was used to replace the second allele. **(B)** Left: DAPI-stained TbTim54 cond. dKO cell line grown in the

presence of tetracycline (+Tet) and 3 or 6 days after the removal of tetracycline from the medium (-Tet). Right: Quantification of kDNA areas in 69 to 90 DAPI-stained cells. The red line indicates the mean of the kDNA areas for each time point. The mean of the control cells (+Tet) was set to 100%. n.s.: not significant ($P$ value >0.05) as calculated by an unpaired two-tailed $t$ test. Numerical data for panel (A) and (B) are available in S1 Data.
(TIF)

**S5 Fig. Changes in mini- and maxicircle levels can be detected by quantitative PCR.** Upper panels: A quantitative PCR-based (qPCR) method was used to detect changes in steady-state levels of total mini- and maxicircles. Total DNA was extracted from uninduced and 2 to 5 days induced TbPam18 and TbPam16 RNAi cell lines. This DNA was used as the template in PCRs amplifying specific mini- and maxicircle regions or the intergenic region of tubulin. PCR products were analyzed on agarose gels. Lower panels: Densitometric quantification of mini- and maxicircle abundance as detected by qPCR. The ratio of the mini- or maxicircle band intensity and the respective control (tubulin band intensity) was normalized (norm.) to the ratios of uninduced cells. Blue (maxicircles) and red (minicircles) bars represent the mean of 3 independent biological replicates. n.s.: not significant, **: $P$ value <0.05, ***: $P$ value <0.005, ****: $P$ value <0.0001, as calculated by an unpaired two-tailed $t$ test. Numerical data are available in S1 Data.
(TIF)

**S6 Fig. Functional analysis of tagged TbPam18 and subcellular localization of TbPam16 variants. (A)** Growth curve of uninduced (-Tet) and induced (+Tet) cells expressing RNAi-resistant (RNAi-res.) TbPam18-HA (left) or myc-TbPam18 (right) in the background of RNAi against the wild type (wt) TbPam18 (TbPam18-HA and myc-TbPam18 exclusive expressors). Insets, top: Immunoblot analysis of whole cell extracts of uninduced (-) and 2 days induced (+) cells, probed for RNAi-res. TbPam18-HA or myc-TbPam18 and EF1a as loading control. Insets, bottom: RT-PCR products of the wt TbPam18 mRNA in uninduced (-) or 2 days induced (+) cells. Tubulin mRNA serves as loading control. **(B)** Upper panels: Immunoblot analysis of total cells (T), digitonin-extracted, mitochondria-enriched (M), and soluble cytosolic (C) fractions of TbPam16-HA, ΔN-TbPam16-HA, and myc-TbPam16 exclusive expressor cell lines. Blots were probed with anti-HA antibodies and antisera against ATOM40 and EF1a, which serve as mitochondrial and cytosolic markers, respectively. Lower panels: digitonin-extracted, crude mitochondrial fractions (M) were subjected to an alkaline carbonate extraction resulting in a pellet enriched in integral membrane proteins (P) and a soluble supernatant fraction (S). Immunoblots were probed with anti-HA and antisera against ATOM40 and cytochrome C (Cyt C), which serve as marker for integral membrane and soluble proteins, respectively. All immunoblots derive from the same gel. **(C)** Quantification of the expression levels in total cells, lanes T in (B), of TbPam16-HA and ΔN-TbPam16-HA normalized to ATOM40. Numerical data for panel (A) and (C) are available in S1 Data.
(TIF)

**S7 Fig. Multiple sequence alignments of Pam18 homologues and subcellular localization of TbPam18 variants. (A)** Sequence alignment of N-terminal regions of Pam18 homologues of 13 representative eukaryotes. **(B)** Sequence alignment of N-terminal regions of Pam18 homologues of 9 representative trypanosomatids. In (A) and (B) Histidine-Proline-Aspartate (HPD) motifs are highlighted in red and Histidine-Serine-Aspartate (HSD) motifs in green. **(C)** Upper panels: Immunoblot analysis of total cells (T), digitonin-extracted mitochondria-enriched (M), and soluble cytosolic (C) fractions of cell lines expressing N-terminally myc-tagged TbPam18 or Tb/ScTbPam18. Blots were probed with anti-myc antibodies and antisera

against ATOM40 and EF1a, which serve as mitochondrial and cytosolic markers, respectively. Lower panels: digitonin-extracted crude mitochondrial fractions (M) were subjected to an alkaline carbonate extraction resulting in a pellet enriched in integral membrane proteins (P) and a soluble supernatant fraction (S). Immunoblots were probed with anti-myc and antisera against ATOM40 and Cyt C, which serve as makers for integral membrane and soluble proteins, respectively.
(TIF)

**S8 Fig. AlphaFold prediction of MaRF11/Tb927.11.435.**
(TIF)

**S9 Fig. Effect of MaRF11/Tb927.11.435 RNAi on the mitochondrial proteome.** Mitochondria-enriched fractions of uninduced and 4 days induced MaRF11 RNAi cells were analyzed by SILAC-based quantitative mass spectrometry. The data set was filtered for mitochondrial proteins and the mean $\log_2$ of normalized ratios (induced/uninduced) was plotted against the corresponding negative $\log_{10}$ of adjusted *P* value (limma test). Highlighted are mitochondrial ribosomal proteins (MRPs, red) and components of the oxidative phosphorylation pathway (OXPHOS, green) as well as TbPam16 (pink). The horizontal dotted line in each volcano plot marks an adjusted *P* value of 0.05. The vertical dotted lines indicate a fold-change in protein abundance of ±1.5. The percentages of all detected MRPs or OXPHOS proteins that are depleted more than 1.5-fold is indicated at the top of each panel. The number of all more than 1.5-fold depleted MRPs or OXPHOS proteins and the total number of all detected MRPs or OXPHOS proteins is shown in parentheses. MaRF11/Tb927.11.435 was not detected. Numerical data are available in S3 Table.
(TIF)

**S10 Fig. MaRF11/Tb927.11.435 RNAi does not affect protein subgroups exclusively consisting of nucleus-encoded proteins.** Mitochondria-enriched fractions of uninduced and 4 days induced MaRF11 RNAi cells were analyzed by SILAC-based quantitative mass spectrometry (same data set as in S9 Fig). **(A–F)** The indicated subgroups of mitochondrial proteins are highlighted in the indicated colors. The number of all more than 1.5-fold depleted proteins and the total number of all detected proteins for each subgroup are indicated in parentheses at the top of each panel. Numerical data for panel are available in S3 Table.
(TIF)

**S11 Fig. Correlation analysis of TbPam18, TbPam16, MaRF11/Tb927.11.435, and TbTim17 SILAC RNAi data sets.** The mitochondrial proteins detected in the data sets of the indicated SILAC RNAi analyses were quantitatively compared in a pairs diagram created with R (version 4.2.1). The $\log_2$-normalized foldchanges of all detected mitochondrial proteins of 1 SILAC experiment were compared to each of the other experiments in individual scatter graphs. Yellow clouds indicate the range in which 50% (inner cloud) and 90% (outer cloud) of the data points are located. The correlation between the data sets was calculated using the Spearman's rank correlation algorithm and is indicated as $r_s$. *N* indicates the total number of proteins that could be compared between 2 data sets. The 4 panels along the diagonal axis contain histograms combined with density graphs displaying the overall data point distribution in each experiment. As all 4 data sets have a left-skewed distribution, the Spearman's rank correlation was used as the statistical measure. Numerical data are available in S1 and S3 Tables.
(TIF)

**S12 Fig. Synthetic TbPam18 and TbPam16 genes.** RNAi-resistant (RNAi-res.) TbPam18 and TbPam16 DNA sequences in black with changed nucleotides highlighted in red. In the

ΔN-TbPam18 and ΔN-TbPam16 constructs, the first 81 and 156 nucleotides, respectively, were replaced by the first 45 nucleotides of TbmHsp60, which encode the mitochondrial targeting sequence of the protein (green). To generate the Tb/ScPam18 fusion protein, the first 295 nucleotides of RNAi-res. TbPam18 were fused to the last 213 nucleotides of yeast (Sc) Pam18 (blue). To generate RNAi-res. TbPam18-S98P, the cytosine at position 292 of the nucleotide sequence was exchanged against a thymine (highlighted in yellow).
(TIF)

**S1 Table. SILAC RNAi TbPam18 and TbPam16.**
(XLSX)

**S2 Table. SILAC CoIPs TbPam16.**
(XLSX)

**S3 Table. TbMaRF11 SILAC RNAi.**
(XLSX)

**S1 Data. Numerical data for all graphs presented in the study.**
(XLSX)

**S1 Raw Images. Original images for all gels and blots.**
(PDF)

## Acknowledgments

We thank Julian Bender and Johannes Zimmermann for assistance in bioinformatics data analysis.

## Author Contributions

**Conceptualization:** Corinne von Känel, Philip Stettler, Bettina Warscheid, André Schneider.

**Data curation:** Silke Oeljeklaus.

**Formal analysis:** Corinne von Känel, Philip Stettler, Carmela Esposito, Stephan Berger, Simona Amodeo, Silke Oeljeklaus, Salvatore Calderaro, Bettina Warscheid, André Schneider.

**Funding acquisition:** Bettina Warscheid, André Schneider.

**Investigation:** Corinne von Känel, Philip Stettler, Carmela Esposito, Stephan Berger, Simona Amodeo, Silke Oeljeklaus, Salvatore Calderaro, Ignacio M. Durante, Vendula Rašková.

**Methodology:** Corinne von Känel.

**Project administration:** Bettina Warscheid, André Schneider.

**Supervision:** Bettina Warscheid, André Schneider.

**Validation:** Corinne von Känel, Philip Stettler.

**Visualization:** Corinne von Känel, Philip Stettler, Silke Oeljeklaus, André Schneider.

**Writing – original draft:** Corinne von Känel, Philip Stettler, André Schneider.

**Writing – review & editing:** Corinne von Känel, Philip Stettler, Carmela Esposito, Stephan Berger, Simona Amodeo, Silke Oeljeklaus, Salvatore Calderaro, Ignacio M. Durante, Vendula Rašková, Bettina Warscheid, André Schneider.

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
