## [Editor Report · Decision Letter 0]

24 Nov 2023

Dear André, 

Thank you very much for submitting your manuscript entitled "Evolutionary repurposing of trypanosomal Pam18 and Pam16 reveals a new regulatory circuit for mitochondrial genome replication" for consideration as a Research Article by PLOS Biology. Please accept my apologies for the delay in getting back to you with feedback this week.

Your manuscript has now been evaluated by the PLOS Biology editorial staff and I am writing to let you know that we would like to send your submission out for external peer review.

Once your full submission is complete, your paper will undergo a series of checks in preparation for peer review. After your manuscript has passed the checks it will be sent out for review. To provide the metadata for your submission, please Login to Editorial Manager (https://www.editorialmanager.com/pbiology) within two working days, i.e. by Nov 26 2023 11:59PM.

Kind regards,

Richard

Richard Hodge, PhD

rhodge@plos.org

PLOS

---

## [Decision Letter · Decision Letter 1]

5 Jan 2024

Dear Andre,

Thank you for your patience while your manuscript "Evolutionary repurposing of trypanosomal Pam18 and Pam16 reveals a new regulatory circuit for mitochondrial genome replication" was peer-reviewed at PLOS Biology, which then caught up with the Christmas holidays. I am contacting you in the absence of my colleague Richard Hodge from the office to avoid any further loss of time. Your manuscript has now been evaluated by the PLOS Biology editors, an Academic Editor with relevant expertise, and by four independent reviewers.

As you will see, the reviewers are generally in agreement that the study is interesting and well-conducted. However, they raise overlapping concerns that the model presented regarding the role of TbPAM16/18 in kDNA replication is too speculative at this stage and direct mechanistic evidence for the claims is lacking. Several experiments are proposed, including a control experiment to account for the potential indirect effects of altered import and direct biochemical evidence that TbMaRF11 acts in kDNA replication. Based on the reviewer comments and following discussion with the Academic Editor, it is clear that a substantial amount of work would be required to meet address the concerns raised and make the study appropriate for PLOS Biology. However, given our and the reviewer interest in your study, we are happy to invite a thorough revision of the study that addresses all the reviewers' comments. 

Specifically, we consider that the mechanistic aspects of the model should be addressed as the reviewers request. We also agree with reviewer 4 that the lack of growth inhibition following RNAi of the PAM 16/18 in bloodstream forms is surprising, but not the major weakness of the paper which is focussed on a thorough mechanistic investigation of the procyclic forms. In some ways the blood stage forms are a distraction and the revision might be better spent confirming your model in procyclics. Imaging and colocalization as suggested by reviewers 1 and 3 would be useful. We share reviewer 2’s reservations about the presentation of the silac data, and think the data should be presented differently (more details about this at the end of the letter in "comments from the Academic Editor"). Reviewer 2's request of an analysis of maxi and minicircles in bloodstream forms following RNAi seems reasonable, although perhaps not strictly necessary to prove your model in procyclics. We strongly agree with reviewer 4 that the evidence is consistent with the proposed model, but also with an indirect effect due to altered transport of mitochondrial proteins required for replication. Using the silac data to select a set of depleted proteins is a good suggestion for the proposed control experiment.

Given the extent of revision that would be needed, we cannot make a decision about publication until we have seen the revised manuscript and your response to the reviewers' comments. Your revised manuscript would need to be seen by the reviewers again, but please note that we would not engage them unless their main concerns have been addressed. We appreciate that these requests represent a great deal of extra work, and we are willing to relax our standard revision time to allow you 6 months to revise your study. Please email us (plosbiology@plos.org) if you have any questions or concerns, or envision needing a (short) extension.

**IMPORTANT - SUBMITTING YOUR REVISION**

*Resubmission Checklist*

*Published Peer Review*

*PLOS Data Policy*

*Blot and Gel Data Policy*

Sincerely,

Nonia

Nonia Pariente, 

Editor in Chief

PLOS Biology

npariente@plos.org

on behalf of

Richard

Richard Hodge, 

Senior Editor

PLOS Biology

rhodge@plos.org

REVIEWS:

Reviewer #1: 

In this paper the authors show that in T. brucei procyclic forms, depleting PAM16 or PAM18 causes loss of maxicircles and release of minicircles from the kDNA network - presumably because they can't re-attach after replication. Oddly, the complex appears not to be required in bloodstream forms although there is no evidence that kDNA replication, per se, is any different from replication in procyclic forms. They demonstrate that the PAM16-PAM18 dimer is associated with a basic protein, MaRF11, depletion of which gives the same phenotype. There is also an association, of unknown significance, with the coenzyme Q complex. The transmembrane domains of PAM16 and PAM18 are required for function. The authors conclude: "the two proteins have switched their function during evolution, from protein import to mitochondrial genome replication." Since the functions of these proteins have not been examined in many different organisms, one could also consider the possibility that it's the other way around - that "the two proteins have switched their function during evolution, from mitochondrial genome replication to protein import." 

The data are generally solid and support the direct conclusions. Weaknesses are:

a) The precise roles of PAM16/PAM18/MaRF11 in kDNA network replication remain a mystery. The model in Fig 10, suggesting a role in nuclear-mitochondrial communication in kDNA replication control is wholly speculative. I can only think of one simple experiment to examine this - immunofluorescence using the Pam16-myc cell line to see whether the complex is concentrated near the kinetoplast. (If so, this might support a mechanistic role.)

b) The lack of an effect of eliminating the complex in bloodstream forms is unexplained. The relevant parasgraph in the discussion is unilluminating, saying merely (I paraphrase) that "control is different". However I thought (maybe I'm wrong) that the major difference between bloodstream and procyclic form cycles are more in control of cytokinesis than of kDNA replication: procyclics can divide to make anucleate "zoids" containing just the kinetoplast, whereas bloodstream forms can't. 

Suggested changes to the text:

The Introduction needs simple diagrams of the import complexes and the kDNA.

The Introduction needs to explain in detail the evidence that TbPam18 and TbPam16 are not required for protein import. This is really central to the start of the paper. (It's not possible to test mitochondrial import in dead parasites...) 

The SILAC results need more discussion since most of the decreased proteins aren't mentioned at all. IN the supplementary tables, please include the most recent annotation, at least for the most reliably and strongly reduced proteins. It's notable that (applying strict filters) most of the decreased proteins are mitochondrial, and include various COX components. Why weren't the Pam proteins found in the RNAi SILAC? The supplementary tables need legends. (Easiest to put it on a separate sheet.)

The link with coenzyme Q needs to be discussed in a bit more detail, at least with a more precise description of what the Q complex does and also what, if anything, it might do in bloodstream forms. (Saying that CoQ is involved in "ubiquinone synthesis" is, to me at least, misleading - it doesn't make the whole molecule.) The fact that RNAi against COQ8 had no effect on kDNA is not completely conclusive, since the effect on protein levels was not measured. Could Pam18/16 nevertheless in some way be linked to the ubiquinone/ubiquinol redox cycle? 

Minor details (typos, grammar etc.)

Abstract:

I don't know what the authors mean by "MaRF11 directly mediates". It's the verb "mediates" that is the problem. For something to mediate we need to know what two things the protein is mediating between. Can the authors either be more specific, or choose a different wording?

Introduction

"The single mitochondrion of trypanosomes contains a single unit genome" .. This wording ("single unit") suggests that there is only a single genome, so it's confusing. How about turning the sentence around?

"The single mitochondrion of trypanosomes contains the most complex mitochondrial genome known in nature, called kinetoplast DNA."

"(ca. 5000 copies, 1 kb each) that are arranged" - "that" defines, "which" describes. So it should be "Which".

"The kDNA network consists to 90% of minicircles," Can't say "consist to". Try e.g. "minicircles comprise 90% of The kDNA network".

Results:

SILAC - I assume that both Pam16 and Pam18 were found in the uninduced controls? If so, need to say this. Or if not, why not?

The percentages of reduced proteins are given so as to artificially inflate accuracy - there are fewer than 100 mitoribosomal or oxphos proteins, so giving percentages with decimal places makes no sense (you;d need 1000 to start with), Why not give the raw numbers instead.

A general problem throughout the text: you describe what experiments were done in the past tense then suddenly shift - even mid-sentence - and give the results in the present tense. If experiments were done in the past, then the results were seen in the past too. Paragraph beginning: "To study the effects on the kDNA in more detail, we performed Southern blot analysis.." suddenly changes tense in the middle (paraphrase - "DNA was extracted and maxicircles are decreased.").Other examples: "In the premix experiment, TbPam16 has been enriched...demonstrating that both CoIPs were successful." " This would be "DNA was extracted and maxicircles were decreased." and "TbPam16 was enriched...demonstrating that both CoIPs were successful." Similarly, "the amount of detected minicircles completely shifts to the supernatant" would be better as ""the detected minicircles completely shifted to the supernatant". Similarly "increase in the levels of MaRF11 for up to six days after induction IS observed, whereas the levels of TbPam16 and EF1a REMAINED constant" - change "is" to "was". Many more examples.

"we conclude that the function of TbPam18 is also redundant in the BSF." - better to add the word "probably" here (probably redundant).

For the Pam18 complementation, it's necessary to say that you do not know how much of the protein was produced in the complemented line, so the conclusion regarding the need for the TM domain has to be tentative. It's also necessary to say that the amounts of tagged Pam16s, relative to the amount of native protein, are unknown. Is the -TM version ot Pam16 in the mitochondrion?

"the same methods that were applied for TbPam16 and TbPam18" should either be "the same methods as were applied for TbPam16 and TbPam18" or simply ""the methods that were applied for TbPam16 and TbPam18".

"One way of how TbPam18 and TbPam16 could regulate.." "of how" doesn't work. try replacing with "in which" or simply "that".

"degradation, reminiscent to TbPIF2" - should be "reminiscent of TbPIF2"

"a parallel reduction on the levels of .." should be "reduction in ..."

Reviewer #2: 

The manuscript submitted by Känel et al., titled " Evolutionary repurposing of trypanosomal Pam18 and Pam16 reveals a new regulatory circuit for mitochondrial genome replication" presents a well-executed study investigating the functions of TbPam18 and TbPam16 in trypanosomes. The authors effectively employ SILAC-based quantitative mass spectrometry to analyze RNAi-induced cell lines and uncover a significant impact on mitochondrial proteins, particularly mitoribosomal proteins (MRPs) and components of the oxidative phosphorylation (OXPHOS) pathway. The results, supported by various experimental techniques, reveal a link between the ablation of TbPam18 or TbPam16 and the depletion of maxicircles in kDNA, showing the essential role of these proteins in mitochondrial function. The manuscript effectively connects the observed effects on the mitochondrial proteome to the potential influence on maxicircle stability, showcasing a comprehensive understanding of the interplay between these integral IM proteins and mitochondrial function in trypanosomes. Immunofluorescence and Southern blotting is consistent with a role for TbPam16/18 in reattachment of minicircles. The identification of novel interaction partners Tb927.11.435 and TbCOQ8, and the involvement of the mitochondrial proteomsome further informs the mechanism by which TbPam16/18 impact kDNA replication and opens paths for further research. The clarity of presentation, coupled with well-designed figures and comprehensive discussions, contributes to the overall strength of the manuscript.

Overall, the experiments are well-executed and support the conclusions, thereby contributing significantly to the understanding of TbPam18 and TbPam16 functions and the mechanism of kDNA replication in Trypanosoma brucei. The major weakness lies in the studies on bloodstream forms (BF), which as presented are not definitive; additional Southern blot or qRT-PCR experiments should be performed to strengthen the conclusion that TbPam16/18 do not perform the same function in BF as they do in procyclic forms. Additionally, addressing the below comments would enrich the manuscript and provide a more comprehensive picture of the functional network involving TbPam18 and TbPam16 in trypanosomes throughout the life cycle.

Major comments (needs to be done)

1) With regard to BF parasites (Fig. 4), the authors conclude that Pam16/18 do not play the same role in BF as they do in procyclics. However, it remains possible that a small kDNA replication defect might not lead to a growth defect in BF. The authors need to at least analyze maxicircle abundance and minicircle release (as in Fig. 3) for the two BF cell lines to support the lack of any effect.

Minor comments (would strengthen the manuscript)

1) In Fig. 1A, the authors show the absence of TbPam16 in the TbPam18 RNAi line, suggesting a dependency of TbPam16 stability on TbPam18. However, the manuscript falls short in illustrating the impact of TbPam16 RNAi on TbPam18 stability. Creation of a cell line with tagged TbPam18 and TbPam16 RNAi would round out this story.

2) In the SILAC experiment in Fig. 1B, the authors state that "…58.6% and 62.5% were depleted more that 1.5 fold…". Are these the percentages that also pass the p<0.05 cutoff or the total number regardless of significance? Please clarify.

3) In Fig. 10, the model lacks influence as it primarily recapitulates the results without explaining the sequential order or direction of action of the proteins in wild type cells. A clearer demonstration of the order of events and the directionality involved is essential for a more compelling model.

4) Page 15 line 2; The sentence beginning "Whereas….." is not a complete sentence. please correct the language.

Reviewer #3: 

In this carefully executed study, von Känel et al. show that the inner membrane proteins TbPam18 and TbPam16, whose homologs regulate the pre-sequence translocase in most eucaryotes, are essential for maxicircle replication in the kinetoplast genome of procyclic (insect) developmental form of T. brucei. This is an intriguing connection, perhaps even a physical one, with the mitochondrial nucleoid as it connects through both mitochondrial membranes with the basal body of flagellum. It follows, that TbPam18 and TbPam16 anchoring in the inner membrane is critical for their function, which links maxicircle replication to the inner membrane. Along with their interactor discovered in this study, a positively charged protein MaRF11, these proteins join an exclusive club of factors essential for maxicircle, but not minicircle, replication. It is unclear whether these findings can be extended to the bloodstream form but the differential impact of a genetic ablation or a mutation in procyclic and bloodstream forms is not unprecedented. Although the experimental data are of high quality and generally support the most immediate conclusions, the model for a membrane-bound circuit that controls the replication (Fig. 10) is premature and too speculative. As written, this is a nice discovery work with several promising, but largely unsubstantiated mechanistic pitches. The role of TbHslVU in regulating MaRF11 level would be one example that needs to be firmly established. Finally, the authors may wish to consider analyzing submitochondrial localization and/or colocalization of TbPam18 TbPam16, MARF11. If their logic holds truth, it seems that these proteins should be concentrated in the IM adjacent to TAC/ basal body rather than distributed uniformly through the mitochondrial contour, as many IM proteins do. 

Reviewer #4: 

This is a very interesting, very well executed and described paper that explores the functions of the inner mitochondrial proteins TbPAM16 and TbPAM18 in T. brucei. Understanding these functions is of considerable interest because previous work has shown that, despite being orthologues of proteins in other eukaryotes, these factors are not part of the T. brucei PAM translocase, where their functions have been assumed by TbPAM27. Thus, the roles of TbPAM16 and TbPAM18 are mysterious. Through a very nice set of experiments, the authors demonstrate that loss of either protein leads to maxicircle kDNA (mitochondrial genome) loss and release of minicircles, and that their essentiality in procyclic form (PCF) parasites is dependent on their transmembrane domain. Moreover, they reveal interaction with several proteins, including TbMaRF11, whose loss also results in PCF-specific growth defects and kDNA loss, and whose abundance is under the control of the mitochondrial protease TbHsIVU.

From the above, exciting data the authors derive a model (Figure 10) in which TbPAM16, TbPAM18 and TbMaRF11 act in kDNA replication: TbMaRF11 acting directly, and TbPAM16/TbPAM18 by controlling TbMaRF11 proteosome-mediated degradation. While such a model would be very novel and interesting, unfortunately their data do not go far enough to prove it and it is therefore overly speculative. There are two main problems that need to addressed.

First, it is clear that TbPAM16 and TbPAM18 are mitochondrial membrane proteins, and the authors' previous work indicates they are not part of a PAM translocase. However, is it truly unique that loss of TbPAM16 and TbPAM18, amongst membrane proteins, results in loss or perturbation of kDNA? A control experiment is needed to show that loss of such proteins does NOT result in kDNA perturbation. It remains possible that TbPAM16 and TbPAM18 contribute in some way to protein import into the mitochondria and, indeed, the authors' SILAC proteome analyses (Fig.1) indicate changes in the abundance of many mitochondrial proteins. It is therefore possible that some of these proteins include factors needed for kDNA replication and the effects documented are an indirect result of altered import. 

Second, the authors propose (in many places, including Fig.10) that TbMaRF11 acts DIRECTLY in kDNA replication. This role and connection to TbPAM16 and TbPAM18 is crucial for the model proposed, but no evidence for a function of TbMaRF11 in kDNA replication, or indeed for localisation to the kDNA is provided. Without any such biochemical or imaging analysis, we cannot discount an indirect role, such as in some form of import, acting with TbPAM16 and TbPAM18.

Minor issues:

1. Page 6, the RNAi experiments do not allow us to conclude that TbPam16 and TbPam18 'form a heterodimer as in yeast'.

2. Page 6. The statement, 'Taken together, the SILAC RNAi results presented here indicate that the ablations of TbPam18 and TbPam16 could affect the kDNA' is premature; this conclusion can only be reached by experiments detailed later.

3. Page 8, 'we conclude that the function of TbPam18 is also redundant in the BSF'; why do they reach this conclusion, given the inability to generate a null mutant?

4. Page 11, The statement 'Furthermore, this also holds true for RNAi-induced knockdown of MaRF11 in BSF NYsm cells, which does not cause a change in the growth rate', should be reworded as it appears to infer that BSF RNAi results in kDNA loss, which has not been documented. 

5. Do the essentially negative results in the section 'The J-domain of yeast Pam18 cannot complement the loss of the TbPam18 J-like domain' add anything to the paper?

Additional comments from the Academic Editor

I find the silac experiment a bit difficult to interpret because it seems to lack a control analysis, ie its a mitochondrial enriched fraction and the two sets of proteins encoded on kDNA are depleted but they need to be compared to a set of mitochondrial proteins not encoded on kDNA to conclude that the kDNA encoded genesets are depleted above background depletion of mitochondrial proteins.

In general abbreviations used infrequently should be replaced with expanded text to aid readibility, there is no word limit in PLOS Biology so no need for abbreviations. Fig 2 a and b tet (d) should be replaced with days as in the graphs. further examples include PCF, BSF, OXPHOS, PAM, IM

Page 8 “Integral membrane localization of TbPam18 and TbPam16 is functionally relevant”. These experiments need to be more clearly explained, preferably in results but could also be in the methods. I assume the tet induces the RNAi but the complementation is also simultaneously induced, it was unclear to me why the complementation was induced by tet instead of being constitutively expressed from the puromycin maintained ectopic plasmid.

Page 12 “Moreover, expression of HA-tagged MaRF11 in the induced TbPam16 RNAi cell line causes a parallel reduction on the levels of TbPam16 and MaRF11, respectively (Fig. 9D).”

I think this sentence is incorrect, the pam16 is reduced because of RNAi not because tagged marf11 is expressed.

---

## [Editor Report · Decision Letter 2]

18 Jul 2024

Dear André,

Thank you for your patience while we considered your revised manuscript "Evolutionary repurposing of trypanosomal Pam18 and Pam16 reveals a new regulatory circuit for mitochondrial genome replication" for publication as a Research Article at PLOS Biology. This revised version of your manuscript has been evaluated by the PLOS Biology editors and the Academic Editor.

Based on our Academic Editor's assessment of your revision, I am pleased to say that we are likely to accept this manuscript for publication, provided you satisfactorily address the following data and other policy-related requests that I have provided below (A-G):

(A) We would like to suggest the following modification to the title:

“Pam16 and Pam18 were repurposed during Trypanosoma brucei evolution to regulate the replication of mitochondrial DNA”

(B) You may be aware of the PLOS Data Policy, which requires that all data be made available without restriction: http://journals.plos.org/plosbiology/s/data-availability. For more information, please also see this editorial: http://dx.doi.org/10.1371/journal.pbio.1001797

-Supplementary files (e.g., excel). Please ensure that all data files are uploaded as 'Supporting Information' and are invariably referred to (in the manuscript, figure legends, and the Description field when uploading your files) using the following format verbatim: S1 Data, S2 Data, etc. Multiple panels of a single or even several figures can be included as multiple sheets in one excel file that is saved using exactly the following convention: S1_Data.xlsx (using an underscore).

-Deposition in a publicly available repository. Please also provide the accession code or a reviewer link so that we may view your data before publication. 

Figure 1B, 2A-B, 4A-D, 5B-D, 6B-C, 7A-B, 8A-B, 8D, 9A, S2, S3, S4A, S4C, S5, S6A, S6C, S9, S10, S11

(C) Thank you for already depositing the SILAC data in the ProteomeXchange database via the PRIDE repository. However, the datasets did not appear when we searched for them in the repository. We ask that you please ensure that the proteomics data is made publicly available at this stage before publication.

(D) Please also ensure that each of the relevant figure legends in your manuscript include information on *WHERE THE UNDERLYING DATA CAN BE FOUND*, and ensure your supplemental data file/s has a legend.

(E) We require the original, uncropped and minimally adjusted images supporting all blot and gel results reported in the following Figures:

Figure 1A, 2B, 3A, 4A-B, 4D, 5B-D, 6B-C, 7B, 8B-D, 9C, S4A, S5, S6B, S7C

We will require these files before a manuscript can be accepted so please prepare and upload them now. Please carefully read our guidelines for how to prepare and upload this data: https://journals.plos.org/plosbiology/s/figures#loc-blot-and-gel-reporting-requirements

(F) Please ensure that your Data Statement in the submission system accurately describes where your data can be found and is in final format, as it will be published as written there. 

(G) Per journal policy, if you have generated any custom code during the course of this investigation, please make it available without restrictions. Please ensure that the code is sufficiently well documented and reusable, and that your Data Statement in the Editorial Manager submission system accurately describes where your code can be found. 

We expect to receive your revised manuscript within two weeks. 

*Published Peer Review History*

*Press*

Best wishes,

Richard

Richard Hodge, PhD

rhodge@plos.org

PLOS

---

## [Editor Report · Decision Letter 3]

23 Jul 2024

Dear André,

On behalf of my colleagues and the Academic Editor, Michael Duffy, I am pleased to say that we can accept your manuscript for publication, provided you address any remaining formatting and reporting issues. These will be detailed in an email you should receive within 2-3 business days from our colleagues in the journal operations team; no action is required from you until then. Please note that we will not be able to formally accept your manuscript and schedule it for publication until you have completed any requested changes.

PRESS

Best wishes, 

Richard

Richard Hodge, PhD

rhodge@plos.org

PLOS
